# Lipid vesicles chaperone an encapsulated RNA aptamer

Ranajay Saha [1], Samuel Verbanic [2] & Irene A. Chen[1,2]

The organization of molecules into cells is believed to have been critical for the emergence of living systems. Early protocells likely consisted of RNA functioning inside vesicles made of simple lipids. However, little is known about how encapsulation would affect the activity and folding of RNA. Here we find that confinement of the malachite green RNA aptamer inside fatty acid vesicles increases binding affinity and locally stabilizes the bound conformation of the RNA. The vesicle effectively 'chaperones' the aptamer, consistent with an excluded volume mechanism due to confinement. Protocellular organization thereby leads to a direct benefit for the RNA. Coupled with previously described mechanisms by which encapsulated RNA aids membrane growth, this effect illustrates how the membrane and RNA might cooperate for mutual benefit. Encapsulation could thus increase RNA fitness and the likelihood that functional sequences would emerge during the origin of life.

[1] Department of Chemistry and Biochemistry, University of California, Santa Barbara, CA 93106, USA. [2] Program in Biomolecular Sciences and Engineering, University of California, Santa Barbara, CA 93106, USA. Correspondence and requests for materials should be addressed to I.A.C. (email: chen@chem.ucsb.edu)

The ability of RNA to act as both an informational and catalytic molecule suggests that a simple living metabolism may require only RNA, which is of high interest for the origin of life as well as for the construction of minimal synthetic cells[1,2]. Encapsulation into primitive cells (protocells) is believed to be critical for the early evolution of life, since this spatial organization prevents dilution of interacting components and prevents parasites from sabotaging the evolution of ribozymes[3,4] with enhanced replication[3,5–7]. Indeed, compartmentalization has become an important strategy for experimental evolution of catalytic nucleic acids[8] and replicases[9]. Moreover, creating a cellular system with emergent properties requires cooperative interactions between the genome and membrane[10,11]. Some mechanisms have been described in which membrane vesicles derive a 'benefit' from encapsulating RNA. In particular, RNA-containing vesicles acquire a growth advantage over empty vesicles[12], the binding of RNA bases can stabilize vesicles against salts[13], ribozyme-catalyzed peptide synthesis might have helped protocells to divide[14], and ribozyme-catalyzed lipid synthesis would stabilize vesicle membranes[15]. However, little is known about the converse interaction, i.e., whether the RNA might derive a benefit from the surrounding membrane vesicle. Vesicle growth can regulate encapsulated ribozymes by diluting inhibitory sequences[16], and dry–wet cycling with lipids may aid the synthesis of sugar-phosphate backbones for nucleic acids[17,18], but mechanisms by which encapsulation itself could improve RNA function have not yet been described. Studying how encapsulation affects functional RNA (e.g., RNA aptamers) is important for understanding the cooperative interactions that are fundamental to the early evolution of simple cells.

While most studies of RNA have been conducted in dilute aqueous solutions, it is well-known that an encapsulated cellular environment differs in important ways from such solutions. For example, the cellular environment may be characterized by depletion effects, membrane surface interactions, and increased effective concentrations, among other effects[19]. Therefore, there is a need to understand how encapsulation inside simple cells affects RNA function as well as folding, since proper folding is generally required for catalytic or binding activity[20]. Previous studies have focused on the effect of model crowding agents on RNA folding and activity, indicating that these agents can stabilize the folded state[21,22] as well as increase the rate of ribozyme reactions[23,24]. For example, polyethylene glycol (PEG) and Ficoll stabilize the folded state of the Azoarcus group I intron, resulting in increased activity[25–27] which can be attributed to the excluded volume effect. Similarly, the HDV-like ribozyme adopted a more native structure and exhibited greater self-cleavage activity in the presence of PEG 8000 and dextran[28]. However, it should be noted that chemical interactions between a nucleic acid and a specific crowding agent can complicate the effect observed from crowding agents[29].

We hypothesized that confinement inside vesicles could improve RNA aptamer activity. Physical mechanisms that lead to increased activity are of special interest for the prebiotic RNA world, because they may raise the likelihood of emergence of functional RNA. Moreover, early studies suggest that the fitness landscape of functional RNAs consists primarily of evolutionarily isolated peaks with distinct structural motifs linked by generally unfavorable paths of mutation[30,31]. An increase in the number and activity of functional sequences could uncover feasible evolutionary pathways for evolution, effectively improving the ability of natural selection to optimize function in sequence space[5,32,33].

In the present work, we encapsulate the malachite green (MG) RNA aptamer inside model protocells and characterize binding affinity and local RNA folding. While modern cell membranes are primarily composed of phospholipids, such membranes are not

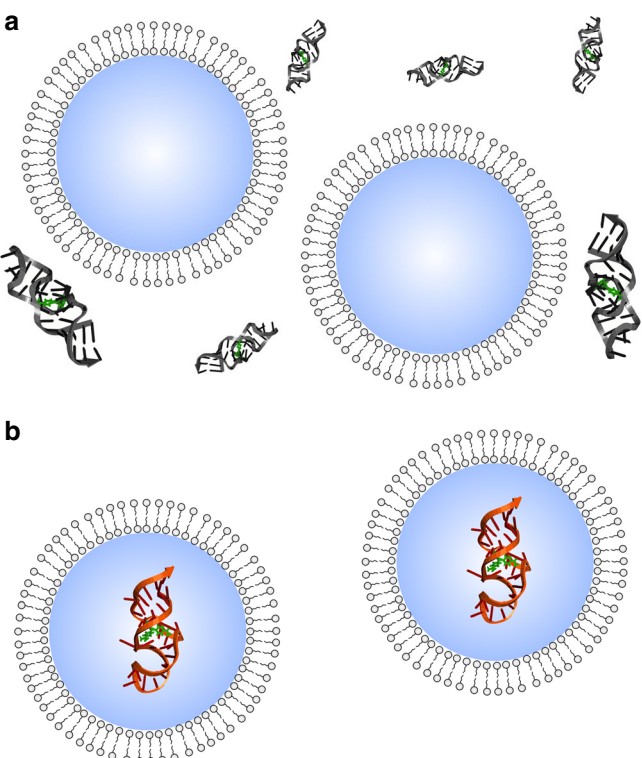

**Fig. 1** Schematic representation of the MG aptamer in different conditions. **a** Exposed to 'empty' vesicles and **b** encapsulated inside vesicles. Comparisons between these two conditions isolate the effect of confinement vs. chemical interaction with the membrane

appropriate for protocells due to their impermeability and slow molecular dynamics, which require protein transporters and flipases to allow internal metabolism and membrane growth. On the other hand, vesicles made from fatty acids are relatively permeable and exhibit faster dynamics, allowing passive diffusion of small molecules (e.g., nucleotides) for RNA replication as well as sustainable membrane growth and division cycles[34–36]. Unlike phospholipids, fatty acids have been synthesized under plausible prebiotic scenarios[37,38]. An earlier study found a modest reduction in self-cleaving activity of a hammerhead ribozyme encapsulated in vesicles assembled from myristoleic acid (MA; C14:1) and a cognate glycerol monoester[39]. However, the mechanism of this effect was not investigated. In particular, it is not known whether the reduction was caused by mere exposure to fatty acid vs. by the confinement of encapsulation. In the present study, in order to isolate the effect of confinement from effects of exposure (i.e., chemical interactions with the membrane), we compare all encapsulation experiments against a control in which aptamer is exposed to vesicles but not encapsulated (Fig. 1). In both situations, the aptamer can interact with vesicle membrane, but in the encapsulated state the aptamer is confined to the vesicle interior, while in the control the aptamer exists outside 'empty' vesicles. This comparison allows us to focus on the following question: In a prebiotic milieu containing lipids, would encapsulation inside a protocell membrane confer any advantage to the RNA compared to the bulk solution?

In principle, reactions involving macromolecular interactions may be subject to changes in effective concentration brought about by encapsulation, and indeed compartmentalization can accelerate ribozyme reactions involving macromolecular interactions through this mechanism[40]. To study mechanisms that are not simply caused by changes in effective concentration, we encapsulated the RNA aptamer for MG, which is permeable and

thus equilibrates across the membrane. Here we show that the affinity and local structural interactions of the MG aptamer are improved by confinement inside vesicles. We discuss the mechanistic interpretation of these results, which suggest an excluded volume effect due to confinement, as well as the evolutionary implications for early life.

## Results

**Vesicles containing myristoleic acid and RNA encapsulation.** Three types of vesicles based on MA were prepared: pure MA, MA with glycerol monomyristolein (GMM) in a 2:1 ratio, and MA with myristoleyl alcohol (MAOH) in a 10:1 ratio[39,41,42]. Vesicle size after extrusion was confirmed by dynamic light scattering (DLS). The hydrodynamic diameter of the vesicles was ~60 nm, consistent with a previous study[41], and was stable after overnight incubation. Separation of vesicles encapsulating RNA and free RNA was achieved by size exclusion chromatography (SEC) with Sepharose 4B resin[34] (Supplementary Fig. 1).

To determine the quantity MG RNA aptamer encapsulated, we measured the amount of RNA encapsulated using quantitative reverse transcription PCR (RT-qPCR) and compared it to the value expected from random encapsulation. In a solution containing 2 mM MA in vesicles (i.e., 6 mM total MA given a critical aggregation concentration (CAC) of 4 mM[41]), given a surface area per MA molecule of $\sim 68 \times 10^{-20}$ m$^2$ (ref. [43]), the total bilayer surface area in solution is ~410 m$^2$ L$^{-1}$. Since a single vesicle of diameter 60 nm has a surface area of ~$10^{-14}$ m$^2$, we estimate a concentration of $\sim 4.1 \times 10^{16}$ vesicles L$^{-1}$. The concentration of encapsulated RNA (after dispersion into bulk solution) was found to be on the order of ~58 nM by RT-qPCR, or $3.5 \times 10^{16}$ molecules L$^{-1}$, so on average we estimate ~0.85 RNA molecules per vesicle. Given the volume of a single vesicle ($\sim 10^{-22}$ m$^3 \approx 10^{-19}$ L), the bulk-equivalent concentration of RNA in the encapsulated volume would be ~14 µM. This value is roughly consistent with the expectation based on random enclosure of RNA during vesicle formation (see Methods). The discrepancy may be due to estimation inaccuracies (e.g., of vesicle diameter, determined by DLS in this study, or of MA surface area, which had been determined in the context of a semicrystalline phospholipid) or from losses caused by charge–charge repulsion between the RNA and MA surface.

**Affinity of the MG aptamer in the absence of vesicles.** Free MG dye has low fluorescence, but the fluorescence increases substantially when bound by the MG aptamer[44]. To determine the dissociation constant ($K_D$) of the MG aptamer in buffer (1 mM Mg-citrate, 10 mM HEPES, 100 mM KCl, 0.2 M bicine, pH 8.5) lacking fatty acid, we measured the fluorescence of the aptamer-MG complex at varying concentrations of MG. The binding curve indicates a $K_D$ of 2.5 ± 0.4 µM (Supplementary Fig. 2a). This value is higher than the $K_D$ value of 800 nM reported previously[45], likely due to the different solute composition of the buffer.

**Affinity of the MG aptamer outside fatty acid vesicles.** The fluorescence response of MG was used to monitor binding activity of the aptamer. We added MG aptamer externally to 'empty' vesicles (not encapsulating RNA) (Fig. 1a); in this case, the aptamer would be exposed to the surfactant effects of the fatty acid without being confined within a compartment. Unencapsulated MG aptamer outside 'empty' vesicles had a higher $K_D$ (e.g., 6.8 ± 1.4 µM for MA vesicles), indicating that the presence of lipid reduced the affinity of the aptamer (Fig. 2a, Supplementary Fig. 2a, Supplementary Table 1). This effect was not sensitive to MA concentrations from 6 to 12 mM (Supplementary Fig. 3), so small variations in MA concentration among different vesicle

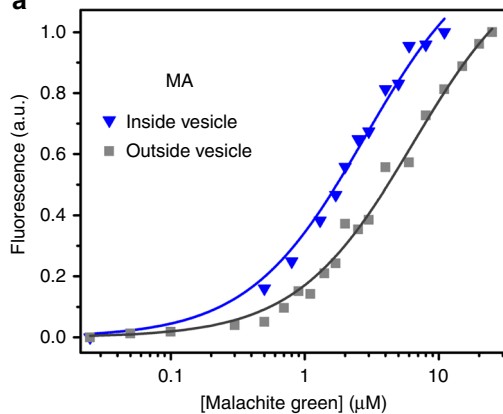

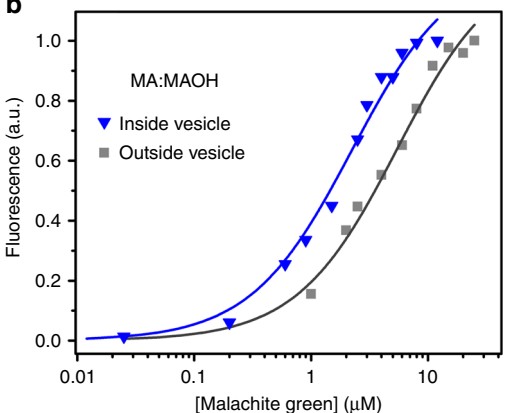

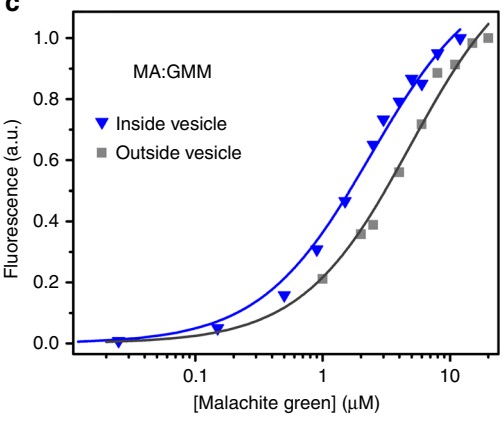

**Fig. 2** MG aptamer affinity in fatty acid vesicles. **a** Encapsulation in MA vesicles (blue triangles) results in increased affinity compared to aptamer exposed to empty vesicles (gray squares). Representative binding curves are shown. Curve fits to the Hill equation are shown as lines (blue: $K_D = 2.5 \pm 0.2$ µM; gray: $K_D = 6.8 \pm 1.4$ µM). Encapsulation in **b** MA:MAOH (blue triangles: $K_D = 2.1 \pm 0.2$ µM; gray square: $K_D = 5.2 \pm 0.1$ µM) or **c** MA:GMM (blue triangles: $K_D = 2.4 \pm 0.1$ µM; gray square: $K_D = 5.1 \pm 0.1$ µM) vesicles shows a similar effect. Values given are mean ± standard deviation

preparations did not influence the observed $K_D$. Similarly reduced affinity was also observed for the MG aptamer exposed to vesicles made from MA:MAOH and MA:GMM compositions (Fig. 2b, c and Supplementary Fig. 2a).

**Affinity of the MG aptamer inside fatty acid vesicles.** We encapsulated the MG aptamer inside MA vesicles (Fig. 1b) and

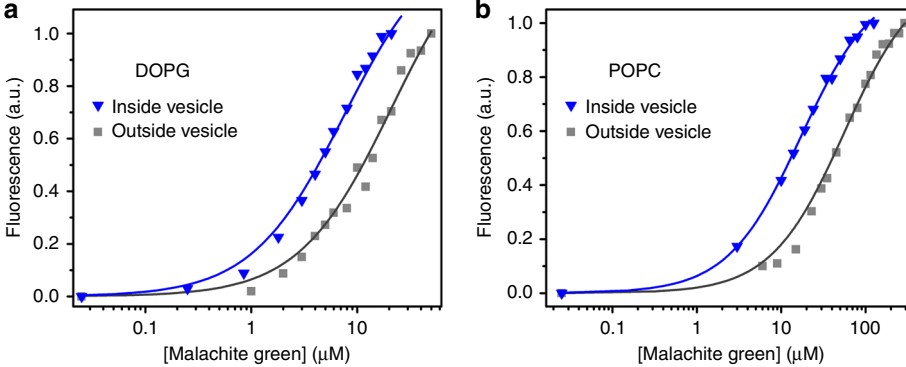

**Fig. 3** MG aptamer affinity in phospholipid vesicles. Encapsulation also increases the binding affinity of MG aptamer to MG in the phospholipid vesicles of DOPG (blue triangles: $K_D = 7.1 \pm 0.2\,\mu\text{M}$; gray square: $K_D = 20 \pm 1.7\,\mu\text{M}$) (**a**) or POPC (blue triangles: $K_D = 18 \pm 1.4\,\mu\text{M}$; gray square: $K_D = 48 \pm 8.6\,\mu\text{M}$) (**b**). Values given are mean ± standard deviation

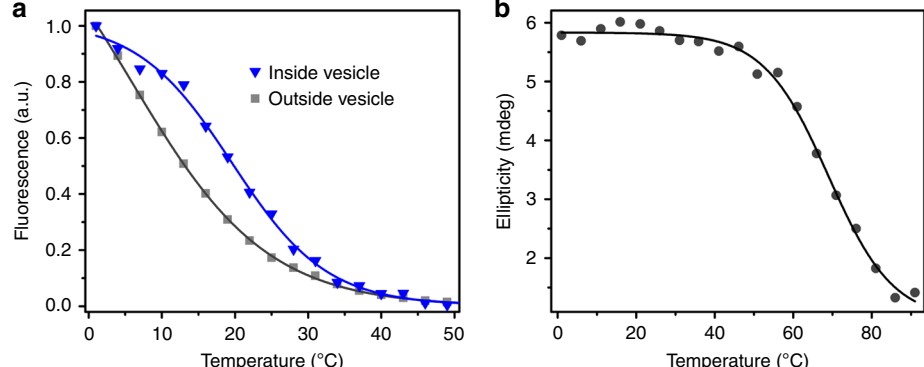

**Fig. 4** Melting transitions of the MG aptamer. **a** Fluorescence response of the MG aptamer with temperature in the presence of saturating MG concentrations ($\sim$3-fold above the $K_D$), for aptamer exposed to empty MA vesicles (gray squares) and aptamer inside MA vesicles (blue triangles). Shown are representative data along with the curve fit to the Boltzmann sigmoidal equation. Note that because the fluorescence of the aptamer exposed to empty vesicles does not show saturation in the measured temperature range, the point of half-maximal measured intensity does not necessarily correspond to the estimated $T_t$. **b** CD response (at 264 nm) of the dye-bound MG aptamer at varying temperatures in buffer without vesicles ($T_m = 71 \pm 2\,°\text{C}$). Values given are mean ± standard deviation

removed unencapsulated RNA by SEC. The collected vesicle fractions were titrated with various known concentrations of MG dye to estimate the $K_D$ value of the encapsulated aptamer. Samples containing vesicles were incubated for several hours to allow MG to equilibrate across the membrane (Supplementary Fig. 4). Affinity was increased if the aptamer was encapsulated inside vesicles ($K_D = 2.5 \pm 0.2\,\mu\text{M}$ for MA vesicles) compared to aptamer exposed to empty vesicles as control (Fig. 2a), indicating that confinement inside vesicles caused a $\sim$3-fold increase in affinity. To mimic more prebiotically plausible protocellular conditions, we repeated these experiments with membranes composed of a mixture of lipids known to increase the stability of fatty acid vesicles[46]. A similar increase in affinity from confinement was found using vesicles composed of MAOH or GMM mixed with MA (Fig. 2b, c).

**Affinity of the MG aptamer with phospholipid vesicles**. To determine whether the effect of encapsulation on affinity depends on the identity of the lipid, we encapsulated the MG aptamer inside DOPG (1,2-dioleoyl-*sn*-glycero-3-phospho-(1′-*rac*-glycerol)) or POPC (1-palmitoyl-2-oleoyl-*sn*-glycero-3-phosphocholine) vesicles. DLS confirmed the formation of both types of vesicles in the same buffer conditions. We first measured the $K_D$ of the aptamer exposed to empty DOPG and POPC vesicles, which gave a $K_D$ of $20 \pm 1.7$ and $48 \pm 8.6\,\mu\text{M}$ (Fig. 3a, b),

respectively. We then encapsulated the aptamer and purified vesicles by SEC. For both DOPG and POPC, the $K_D$ decreased upon encapsulation to $7.1 \pm 0.2$ and $18 \pm 1.4\,\mu\text{M}$, respectively, corresponding to a $\sim$3-fold increase in affinity in both cases (Fig. 3a, b).

**Melting transition of the MG aptamer**. To determine whether the RNA structure was stabilized by encapsulation, we monitored the melting transition of the MG aptamer by monitoring the fluorescence of the complex from 1 to 70 °C. Measurement by fluorescence reports on interactions relevant to ligand binding and not necessarily on the overall fold of the RNA. In the presence of empty MA vesicles (aptamer not encapsulated), a single transition was observed with a transition temperature ($T_t$) of $4.8 \pm 1.1\,°\text{C}$. Encapsulation inside MA vesicles increased the $T_t$ to $20 \pm 0.5\,°\text{C}$ (Fig. 4a). Consistent with the trend in $K_D$, mere exposure to empty vesicles destabilized the aptamer (Supplementary Fig. 5a). To determine whether aptamer–aptamer interactions could induce the stabilization observed with encapsulation, we measured the $T_t$ of non-encapsulated aptamer at high concentration (28 $\mu$M) in the presence of empty MA vesicles. As expected, the $T_t$ was not significantly altered at high concentrations (Supplementary Fig 5b).

To understand whether the transition represented overall folding of the RNA or local interactions in the ligand-binding site,

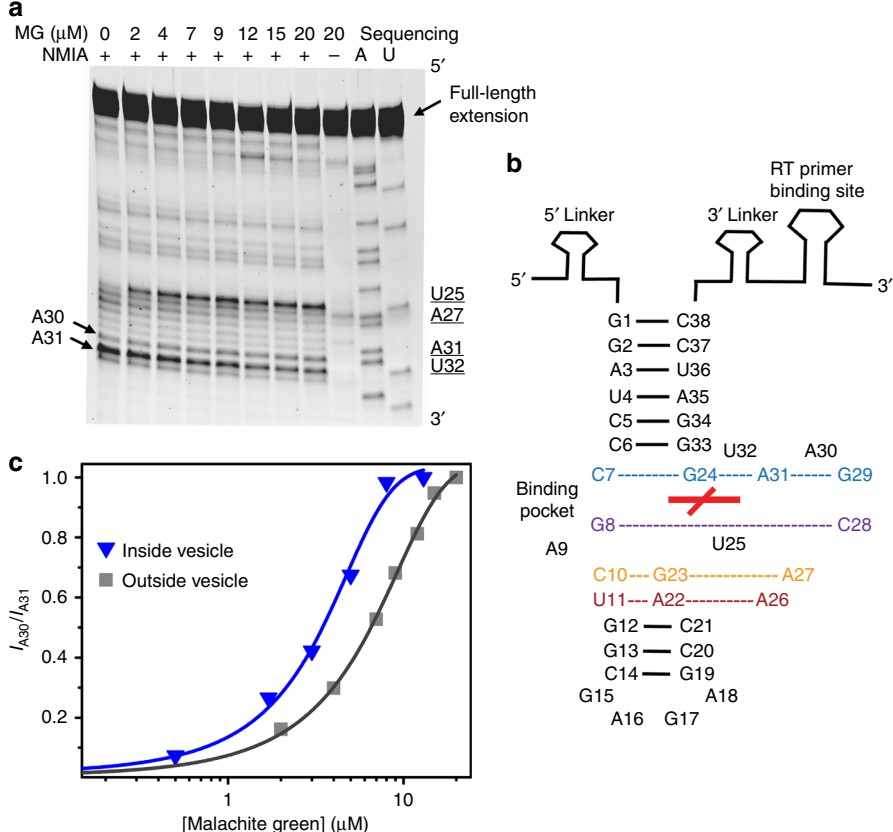

**Fig. 5** Conformational probing of the MG aptamer. **a** Representative SHAPE gel (MG aptamer exposed to empty MA vesicles). The first eight lanes show SHAPE reactions containing NMIA at increasing MG concentrations. The (−) lane shows a negative control containing 20 μM MG but in which no NMIA was added. The two rightmost lanes show A and U sequencing reactions performed using ddTTP and ddATP as chain terminators, respectively. The sequencing lanes give bands that are one nucleotide longer than the corresponding NMIA lanes. **b** Scheme showing the aptamer SHAPE construct and the MG-aptamer structure determined previously[45]. The red cross in the center indicates the position of the MG ligand. Base pairing in stem regions (solid black lines) and other interactions (dotted lines) are shown. **c** Leftward shift in band intensity ratio between A30 to A31 at varying concentrations of MG when the aptamer is encapsulated in MA vesicles (blue triangles) vs. exposed to empty MA vesicles (gray squares). Representative concentration series are shown. Lines show curve fits to the Boltzmann equation

we measured the melting temperature ($T_m$) by circular dichroism (CD) spectroscopy. The $T_m$ of the aptamer in aqueous buffer, without vesicles, monitored by CD, was found to be $71 \pm 2\,°C$ (Fig. 4b), indicating that $T_t$ represents interactions in the ligand-binding site and not global folding or large transitions of secondary structure.

**SHAPE assay of the encapsulated MG aptamer**. To probe aptamer structure inside vesicles vs. exposed to empty vesicles, we assayed the SHAPE (selective 2′-hydroxyl acylation analyzed by primer extension) reactivity of the MG aptamer at different concentrations of the MG dye. The ratio of band intensity of A30 to A31 was found to increase monotonically with increasing MG concentration, consistent with the known participation of these residues in the MG-binding pocket (Fig. 5 and Supplementary Fig. 6)[45]. Therefore, the normalized ratio of band intensity between A30 and A31 ($I_{A30}/I_{A31}$) was used to monitor the conformational transition upon binding to MG. A pronounced leftward shift was observed in $I_{A30}/I_{A31}$ at varying concentrations of MG when the RNA was encapsulated (midpoint at $2.7 \pm 0.5\,μM$) vs. exposed to empty MA vesicles (midpoint at $6.1 \pm 0.7\,μM$; Fig. 5c). The [MG] at the midpoint of each curve agrees with the corresponding $K_D$, supporting the use of these curves to monitor the unbound to bound conformational transition. The results

indicate that the structural transition of the aptamer to the bound state occurs at lower [MG] when the RNA is encapsulated.

**Aptamer affinity in the presence of crowding agents**. To determine whether the effect of encapsulation could be replicated by macromolecular crowding, we determined the $K_D$ of the MG aptamer in the presence of standard crowding agents. Different crowding agents caused qualitatively different effects. Dextran 40,000 (polymer of glucose) decreased the $K_D$ ~2-fold compared to an equivalent amount of glucose (Fig. 6a). The presence of dextran also stabilized the interactions in the ligand-binding region, as demonstrated by an increased $T_t$ (Fig. 6b). However, PEG 8000 increased the $K_D$ with greater effect at higher concentration, and Ficoll PM 70 showed no concentration-dependent effect on $K_D$ or $T_t$ (Supplementary Fig. 7).

**Effect of $Mg^{2+}$ concentration in vesicles**. The MG aptamer is known to be relatively insensitive to $Mg^{2+}$ concentration[47]. Indeed, we observed that the presence or absence of 1 mM $Mg^{2+}$ did not affect the $K_D$ of the MG aptamer inside DOPG vesicles (Fig. 6c). Also, a similar shift in $K_D$ was observed upon encapsulation inside vesicles in the presence or absence of $Mg^{2+}$ (Figs. 3a and 6c). This observation indicates that the effect of encapsulation may be observed in other salt conditions. However,

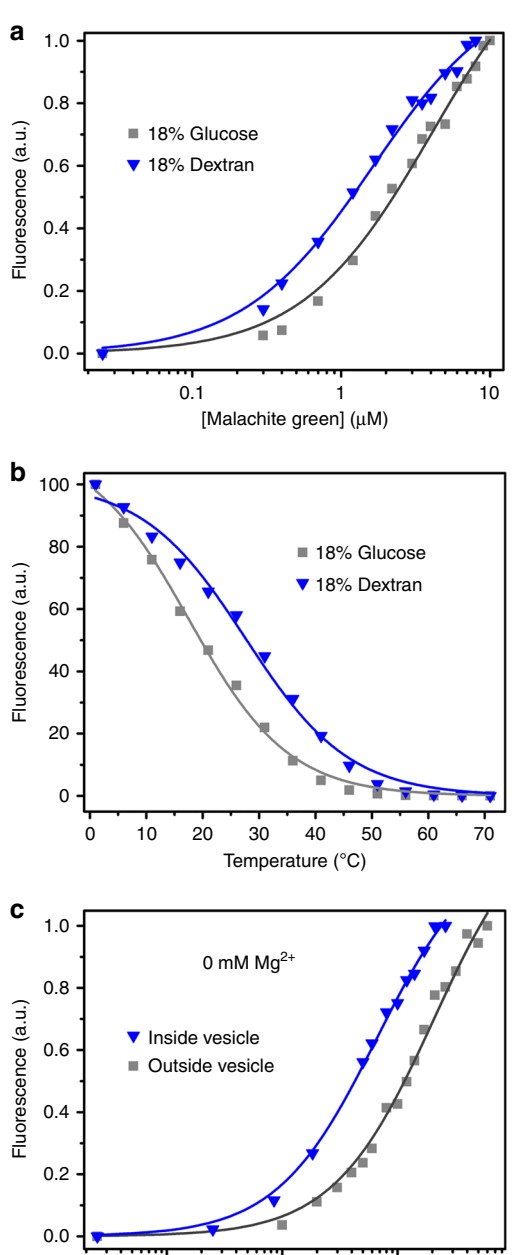

**Fig. 6** MG aptamer affinity in the presence of different solutes. **a** The crowded conditions of 18% dextran (blue triangle, $K_D = 1.8 \pm 0.4\ \mu M$) lowers the $K_D$ compared to 18% glucose (gray square, $K_D = 3.6 \pm 0.4\ \mu M$). **b** The transition temperature ($T_t$) of the MG aptamer (SHAPE construct) also increases in the presence 18% dextran ($T_t = 28 \pm 1.1\ °C$, blue triangles) compared to 18% glucose ($T_t = 18 \pm 0.8\ °C$, gray squares). **c** Binding curve of the MG aptamer in DOPG vesicles without $Mg^{2+}$ in the medium (gray square: outside vesicle, $K_D = 20 \pm 1\ \mu M$; blue triangle: inside vesicle, $K_D = 9.3 \pm 2.4\ \mu M$). Values given are mean ± standard deviation

some functional RNAs require $Mg^{2+}$ for folding and catalysis[48]. To determine whether the internal $Mg^{2+}$ concentration was the same as that in bulk solution, we measured the $Mg^{2+}$ activity inside vesicles by encapsulating the $Mg^{2+}$-sensitive dye mag-fura-2 (Supplementary Fig. 8). Using this assay, the internal $Mg^{2+}$ activity inside vesicles was found to be the same as the bulk concentration of $Mg^{2+}$ (1 mM).

## Discussion

The earliest cells likely existed in an environment that was fairly complex and at least included lipids and various ions. The physico-chemical environment is known to be an important factor in RNA function. For example, in anaerobic conditions RNA can catalyze electron transfer reactions using $Fe^{2+}$ as a cofactor[49], and in icy conditions, a polymerase ribozyme can accurately synthesize an RNA sequence longer than itself[50]. While RNAs would have been exposed to lipids in the prebiotic milieu, compartmentalization inside a protocell was critical for several reasons, including preventing the evolution of genetic parasites. To understand the effect of vesicle encapsulation on RNA activity and folding, crowded conditions can be used as an analog of confinement, but crowding agents show variable effects on RNA depending on the specific crowding agent[29]. Alternatively, RNA could be studied inside living cells[51], but such an environment is much more complicated than a primitive protocell[29,52]. In the model system studied here, we encapsulated the MG aptamer inside vesicles made from prebiotically plausible fatty acids to mimic protocells.

To properly interpret experiments encapsulating RNA in model protocells, it is necessary to first determine what effect simple exposure to the lipid has on RNA folding and function (Fig. 1a). We initially noted that the presence of 'empty' vesicles increased the $K_D$ and decreased the thermal stability of the MG-aptamer complex, suggesting that mere exposure to fatty acid causes some denaturation of the RNA. Interaction between RNA and lipids is believed to be mediated by both electrostatic interactions with the headgroup and hydrophobic interactions to nucleobases[53], and may be enhanced by an ordered bilayer[54]. Electrostatic effects appear to dominate the interaction between fatty acids and minerals[55]. Consistent with the importance of headgroup charge, we found that the zwitterionic lipid POPC decreased aptamer activity more ($K_D = 48 \pm 8.6\ \mu M$) than the negatively charged lipids MA ($K_D = 6.8 \pm 1.4\ \mu M$) and DOPG ($K_D = 20 \pm 1.7\ \mu M$). Anionic detergents (e.g., sodium dodecyl sulfate) have been previously observed to completely inhibit a self-cleaving ribozyme, while non-ionic and zwitterionic detergents substantially enhanced activity[56]. Conversely, the cationic detergent cetyltrimethylammonium bromide (CTAB) has been shown to promote self-cleaving ribozyme reactions[57,58], likely by increasing the rate of association and dissociation, potentially by 3–4 orders of magnitude[59]. Since the prebiotic milieu of protocells would have contained lipids, the comparison between RNA exposed to empty vesicles vs. RNA encapsulated inside vesicles probes how an RNA molecule that became encapsulated in a protocell would differ from an otherwise identical molecule that did not.

We found that the affinity of the MG aptamer increases when encapsulated inside vesicles (Fig. 2a). A similar increase in affinity upon encapsulation was observed for MA, MA:MAOH, MA:GMM, POPC, and DOPG vesicles, suggesting a relatively general phenomenon with respect to lipid composition. On the other hand, a previous study found that the rate of a hammerhead ribozyme was decreased when encapsulated compared to buffer lacking vesicles[39,60]. However, the activity of the ribozyme was not tested when exposed to vesicles without encapsulation, so it is unknown whether the observed effect was due to greater denaturation or to a smaller encapsulation effect compared to the aptamer studied here. It is intriguing to speculate that encapsulation might inhibit a self-cleavage reaction that produces two product molecules but promote an association between two molecules. Further studies are needed to understand how different aptamers or reactions would be affected.

In general, several mechanisms may be considered for the enhancement of the MG aptamer affinity by encapsulation inside

vesicles. First, encapsulation might increase the local concentration of the ligand, as has been observed in other compartmentalized systems[6,40]. However, this mechanism does not apply here, because the small-molecule ligand equilibrates across the membrane within few hours (Supplementary Fig. 4). The encapsulation of a charged polymer does perturb the equilibrium concentration of permeable ions of the opposite charge through the Gibbs–Donnan effect[61]. Qualitatively, this effect would increase the concentration of MG inside the vesicles. However, the size of this effect is very small, corresponding to an [MG] increase of 1.0017-fold inside the vesicles (assuming one RNA molecule encapsulated), well within experimental error. Thus an increased concentration of MG inside the vesicles is unlikely to explain the effect on affinity that was observed. Second, RNA–RNA interactions might be stabilizing and be enhanced inside a minority of the vesicle population that encapsulates multiple RNAs. Chance encapsulation of two RNAs in a vesicle of volume $10^{-19}$ L would correspond to ~30 μM RNA. However, the $T_t$ of the MG aptamer was not increased when measured at this higher concentration. A related possibility would be that membrane-bound MG dye is more favorable for binding by the RNA compared to aqueous MG. However, the observation that confinement in POPC vesicles (which do not bind MG detectably, Supplementary Fig. 9) gives a similar effect as confinement in other membrane types indicates that membrane-bound MG is unlikely to cause the increased affinity. Third, the encapsulated volume might contain a different concentration of $Mg^{2+}$, an important cation for functional RNA. However, the MG aptamer was not sensitive to $[Mg^{2+}]$, and measurement of the internal $[Mg^{2+}]$ indicated no difference from the bulk concentration. This observation also supports the small size of the Gibbs–Donnan effect in this system. A caveat of the measurement of $[Mg^{2+}]$ is that the $K_D$ of mag-fura-2 for $Mg^{2+}$ is 1.5 mM[62], so the assay would not detect binding of $Mg^{2+}$ by lipid through weaker interactions. Fourth, specific chemical interactions between the fatty acid and aptamer might alter RNA activity; this is indeed the case, but the experimental design using RNA exposed to empty fatty acid vesicles as a control clarifies that encapsulation inside vesicles causes increased affinity. In addition, although exposure of the MG aptamer to different lipids results in quantitatively different effects on affinity, encapsulation of the MG aptamer inside vesicles made of different lipids results in a quantitatively similar increase in affinity compared to simple exposure to the corresponding empty vesicles. Fifth, it is possible that the cosolute concentration affects the RNA folding equilibrium. If one considers the membrane itself as a collection of solutes, we can calculate an average value for the fatty acid 'concentration' within the vesicle. For a vesicle of 60 nm outer diameter, MA length ~2 nm and MA surface area ~0.68 $nm^2$, we estimate ~14,000 fatty acid molecules per vesicle interior, and therefore a rough 'concentration' of ~0.3 M for the vesicle interior, so a cosolute effect may be plausible. However, the presence of cosolutes is known to generally decrease the stability of duplexes based on the hydration changes associated with folding[23]. It is possible that a related mechanism is responsible for the decreased affinity observed in the presence of empty vesicles, but the direction of the effect indicates that it is not likely to explain the increased affinity observed inside vesicles.

A sixth possibility is the excluded volume effect, which may alter folding equilibria based on the compactness (and shape) of different states. In essence, the confinement inside a vesicle disfavors extended structures and shifts the equilibrium toward compacted structures. Although this effect is typically examined using crowding agents[21,25], confinement by a membrane should produce a similar effect. Indeed, Dextran 40,000 (polymer of glucose) induced an increase in MG aptamer affinity compared to

an equivalent amount of monomer glucose molecules. However, as seen in other studies[19], the chemical nature of the crowding agent is important, and indeed we observed qualitatively different effects using Dextran, PEG, and Ficoll. Several studies suggest that PEG in particular interacts nonspecifically with nucleic acids and proteins, leading to nontrivial effects[29,33,63,64]. These results indicate that the excluded volume effect is not necessarily the primary driver of the effect of crowding agents on the MG aptamer, and chemical interactions with cosolutes would be an important factor affecting aptamer affinity. As seen by the range of effects we observe in the presence of different crowding agents and lipids (Supplementary Fig. 7 and Supplementary Table 1), cosolutes can have a major effect on the affinity of the MG aptamer. The largest effect observed here was a 20-fold decrease in affinity upon exposure to POPC, and indeed positively charged species can cause nonspecific precipitation and aggregation of RNA. Thus chemical interactions with the RNA could be a dominant influence on aptamer affinity, depending on the solution composition. However, unlike the excluded volume effect of crowding agents, the confinement effect of vesicles can be studied separately from the chemical interactions by experimental comparison with non-encapsulated RNA that is exposed to vesicles (RNA outside vesicles). This comparison indicates a similar increase in affinity (~3-fold) observed for the five lipid compositions tested here. Given this, a potential distinction of confinement inside vesicles compared to crowding agents is the robustness to changes in membrane composition.

If the excluded volume effect is based on a change in equilibrium between folded and unfolded structures, this change should be found when assaying the conformation of the RNA. We found a large decrease in the $T_t$ monitored by fluorescence when RNA was exposed to MA vesicles, indicating decreased stability of interactions relevant to ligand–RNA binding (e.g., contacts between ligand and RNA, or local tertiary contacts in the binding region). A substantial decrease in the transition temperature was also observed in the presence of 18% glucose (34 vs. 18 °C), indicating that this cosolute is also destabilizing. However, both encapsulation and the crowding effect of 18% dextran substantially increased the $T_t$ (Figs. 4a and 6b), suggesting local structural stabilization. This degree of stabilization is similar to previous observations and models. For example, encapsulation of α-lactalbumin in the pores of silica glass was found to increase the melting temperature by 32 °C[65]. Using molecular dynamics simulations, an RNA pseudoknot was shown to be stabilized, relative to the extended hairpin structure, by crowding agents, with an expected melting temperature increase from 78 to 91 °C[66]. A statistical-thermodynamic model of proteins in physiological crowding conditions predicts stabilization of folding transitions by 5–20 °C[67]. It is also possible that the effect of confinement may be greater at certain local features (e.g., a pseudoknot or ligand-binding site) than for the global structure. To probe the structural transition in more detail, we assessed local nucleotide flexibility in the RNA using the SHAPE assay, which measures the relative nucleophilicity of the ribose 2′-OH groups[68]. The MG aptamer is known to change conformation during complex formation ('induced fit')[45,69]. SHAPE assays confirm that a structural transition to the bound conformation occurs at lower [MG] when the aptamer is encapsulated vs. exposed to empty vesicles. The results from SHAPE are consistent with a confinement effect, and demonstrate the utility of SHAPE for encapsulated reactions. These results are consistent with prior work, in which small RNAs were probed by NMR spectroscopy and their structures appeared to be stabilized in reverse micelles[70], likely due to restricted local motion. In our work, we show that confinement (in vesicles) also leads to functional improvement for the MG aptamer. Previous studies of a small

DNA duplex indicated that binding of ethidium bromide was decreased by DNA condensation in reverse micelles[71]. Thus while nonspecific intercalation in DNA may be inhibited by confinement, the specific binding interaction between an RNA aptamer and its target can be enhanced.

A theoretical model of confinement was developed by Zhou and Dill[72], in which a polymer is modeled as existing in either the folded conformation or as a freely jointed chain (unfolded). The effect of confinement on the $\Delta\Delta G$ between the folded and unfolded states can be predicted from this model. Using parameters for ssDNA for the unfolded state (Supplementary Fig. 10), the approximate radius of the folded MG aptamer (1.7 nm[45]), and the inner diameter of the vesicle (52 nm for MA vesicles), the $\Delta\Delta G$ is predicted to be $-0.4kT$ for encapsulation inside MA vesicles (where $k$ is the Boltzmann constant and $T$ is absolute temperature). The three-fold increase in affinity observed in our experiments translates to $\Delta\Delta G = -1.1kT$ stabilization. Given the simplifying assumptions made by the model (e.g., two-state folding neglecting metastable misfolded states; neglect of electrostatic interactions), the experimental observation is in rough agreement with the theoretical prediction. In principle, varying the confinement size and the RNA size might allow further probing of this effect, although the quantitative variation of this effect within the regime of experimentally tractable vesicle sizes and aptamers of typical size is predicted to be small (Supplementary Fig. 10).

RNA is considered to be a crucial molecule for the origin of life. Any general mechanism that could increase the activity of functional RNAs would also increase the number of sequences existing above a critical fitness (e.g., an error threshold), and thus raise the probability that an active sequence would emerge. Encapsulation favors compact structures relative to extended structures, an effect that would be more important for poorly folded RNAs. To the extent that fitness depends on proper folding, encapsulation would also 'flatten' the fitness landscape, permitting greater sequence diversity. The importance of chaperone-like activity has been emphasized in the early co-evolution of ribosomal RNA and protein[73]. Our results suggest that encapsulation could act as a primitive chaperone for RNA, favoring folded structures inside a protocell. An important caveat is that the effect of encapsulation was only studied for one aptamer here, in a limited number of chemical conditions and vesicle compositions. In addition, while this biophysical effect may occur in a modern biological context, it is unclear whether it would be quantitatively important since chemical interactions with the complex cellular or exosomal contents may play a large role in determining RNA folding. Further work would be needed to assess the generality of the effect. While previous studies have demonstrated mechanisms by which RNA activity could contribute to membrane growth and dynamics[12–14], here we show that encapsulation inside a membrane vesicle would have a direct beneficial effect on the function of the RNA, closing the loop to create a mutually beneficial interaction in a simple cell-like system.

## Methods

**Materials**. Myristoleic acid (MA, C14:1), monomyristolein (GMM), and Δ9 myristoleyl alcohol (tetradecenol, MAOH) were purchased from NuChek Prep (Elysian, MN). Phospholipids POPC and DOPG were purchased from Avanti Polar Lipids (Alabaster, AL). Bicine (Alfa Aesar), Sepharose 4B (Sigma-Aldrich), malachite green (MG) chloride (Sigma-Aldrich), Dextran 40,000 (Spectrum Chemical MFG Corp.), PEG 8000 (Promega), Ficoll PM 70 (Sigma-Aldrich), N-methylisatoic anhydride (NMIA) (Molecular Probes), dimethyl sulfoxide (DMSO) (Life Technologies), Mag-Fura-2, tetrapotassium salt (Molecular Probes, Eugene, OR) and GlycoBlue (Invitrogen) were used as received. The MG RNA aptamer (5'-GGAUCCCGACUGGCGAGAGCCAGGUAACGAAUGGAUCC-3') and other oligonucleotides were obtained by chemical synthesis and HPLC-purified by IDT.

The MG aptamer construct for SHAPE study was prepared by in vitro transcription. Other chemicals not mentioned above were purchased from Fisher Scientific.

**Preparation of fatty acid vesicles**. Previous methodology was adopted in order to prepare vesicles from MA using the pH-drop method[41]. Briefly, 3.7 μL of neat MA was mixed with one equivalent of 1 M KOH (15 μL) in water to produce a clear solution of 800 mM MA (micellar). To this solution 120 μL of 1 M bicine and 60 μL of 10× salt mix (100 mM HEPES, 10 mM Mg-Citrate, 1 M KCl, pH 8.5) and water were added to produce a visibly turbid solution of 25 mM MA in 0.2 M bicine, 10 mM HEPES, 1 mM Mg-Citrate, 100 mM KCl, pH 8.5. KOH was used instead of NaOH as vesicles were noted to be less prone to aggregation in the presence of chelated Mg$^{2+}$ ions[74]. After extrusion through 100 nm polycarbonate membrane filters (Whatman) using a Mini-Extruder (Avanti Polar Lipids, Alabaster, AL), the vesicle solutions were tumbled overnight for equilibration before any subsequent experiments. For mixed composition lipid vesicles, the neat oils were mixed by pipetting before dispersion in buffer solutions[39]. The molar ratio of GMM and MAOH to MA was 1:2 and 1:10, respectively.

**Determination of CAC of MA:MAOH vesicles**. The absorbance at 600 nm was monitored with serial dilution using the cuvette-based application of the IMPLEN P300 nanophotometer. Two milliliter of a 4.3 mM vesicle suspension was prepared and the solution was serially diluted with 100–600 μL of buffer solution containing 10 mM HEPES, 1 mM Mg-Citrate, 100 mM KCl, 0.2 M bicine, pH 8.5. The point of inflection of the curve was used to estimate the CAC of the vesicles (Supplementary Fig. 11).

**Preparation of phospholipid vesicles**. To prepare phospholipid vesicles, 12 mM DOPG or 20 mM POPC dissolved in chloroform was dried by rotary evaporation onto a round-bottom flask and resuspended in 0.2 M bicine, 1 mM Mg-Citrate, 100 mM KCl, 10 mM HEPES, pH 8.5. For preparing vesicles encapsulating RNA, the resuspension buffer also included annealed RNA (~40 μM final concentration; see below for annealing procedure). The DOPG suspensions were stirred at 45 °C for 2 h before freeze–thawing five times with dry ice and a water bath at 45 °C. For POPC, the vesicles were freeze–thawed repetitively (10 times) with dry ice and a 25 °C water bath. Vesicles were extruded through 100 nm pores as described above, and equilibrated overnight with end-over-end tumbling.

**DLS measurement**. DLS measurements of the extruded vesicle solutions were performed using the Zetasizer Nano ZSP (Malvern Instruments, UK) at room temperature. Photons were collected at 173° scattering angle and the scattering intensity data were processed using the instrumental software to determine the hydrodynamic size of the vesicles.

**Encapsulation of RNA**. The RNA aptamer solution (~57–88 μM in 50–150 μL of 10 mM Tris-Cl, pH 8.5) was heated to 90 °C for 3 min and annealed by cooling on ice for 15 min. The annealed RNA was mixed with buffer so as to obtain a final concentration of ~33–58 μM RNA in 0.6–2.5 mM Tris, 10 mM HEPES, 1 mM Mg-Citrate, 100 mM KCl, 0.2 M bicine, pH 8.5, after addition of MA. This solution was added to the MA micellar stock (see above) to prepare vesicles encapsulating RNA. The use of citrate-chelated Mg$^{2+}$ (1 mM) instead of MgCl$_2$ reduces the destabilizing effect of Mg$^{2+}$ on fatty acid membranes[74,75]. Vesicles were extruded (see above) and purified from unencapsulated (free) RNAs using a Sepharose 4B size exclusion column[12,34] with a mobile phase consisting of 10 mM HEPES, 1 mM Mg-Citrate, 100 mM KCl, 0.2 mM bicine, pH 8.5, 4 mM MA. The presence of 4 mM MA ensures membrane integrity of the vesicles during column purification and dilution, as the CAC of MA is ~4 mM[41]. For the purification of mixed composition vesicles, the corresponding lipid concentration used during purification was 1 mM for MA:GMM vesicles[12] and 2.8 mM for MA:MAOH vesicles (the CAC of MA: MAOH vesicles was estimated to be ~1.5 to 2 mM; Supplementary Fig. 11). For purification of phospholipid vesicles, the mobile phase did not contain additional lipids since the CAC of phospholipids is much lower (<10$^{-8}$ M[76]) than the lipid concentrations used here.

**Preparation of RNA exposed to empty vesicles**. A 50 μL solution was prepared containing 4 μM RNA aptamer, 6.6 mM Tris, 10 mM HEPES, 1 mM Mg-Citrate, 100 mM KCl, 0.2 M bicine, pH 8.5. This solution was added to 100–150 μL of preformed MA vesicles (25 mM MA, 10 mM HEPES, 1 mM Mg-Citrate, 100 mM KCl, 0.2 M bicine, pH 8.5) and 200–250 μL of buffer (4 mM MA micelles, 10 mM HEPES, 1 mM Mg-Citrate, 100 mM KCl, 0.2 M bicine, pH 8.5) to obtain a solution containing 0.5 μM RNA (non-encapsulated) and 8.8–11 mM MA. For $K_D$ measurements, 20 μL of this solution was added to 20 μL of solution containing 4 mM MA, 10 mM HEPES, 1 mM Mg-Citrate, 100 mM KCl, 0.2 M bicine, pH 8.5, and varying concentrations of MG, and treated as described below. For MA:GMM and MA:MAOH vesicles, the same procedure was followed except that the buffer included the appropriate concentration (≥CAC) of the corresponding lipid solution. For phospholipid vesicles, no additional lipid was added during dilution steps due to the low CAC of these lipids.

**Dissociation constant measurement.** MG was dissolved in 4 mM MA solution containing 10 mM HEPES, 1 mM Mg-Citrate, 100 mM KCl, 0.2 mM bicine, pH 8.5, to obtain a stock solution of 1–80 μM MG. Purified vesicles were incubated with various known concentrations of MG. The mixture was incubated at room temperature overnight, which allowed equilibration of the dye across the membrane. The steady-state fluorescence of the MG aptamers at multiple concentrations of dye ligand was measured using the TECAN M200 Pro plate reader. The MG dye was excited at 617 nm and the emission monitored at 660 nm. Fluorescence intensity was normalized to a minimum of 0 and a maximum of 1. In cases including negatively charged membranes, to account for fluorescence signal arising from MG bound to the membrane (Supplementary Fig. 2b), the background fluorescence ($\lambda = 660$ nm) of a corresponding control sample containing the dye in the presence of the same concentration of vesicles (without aptamer) was subtracted from the fluorescence ($\lambda = 660$ nm) of vesicles with aptamer-bound dye (encapsulated or non-encapsulated). This background correction was not necessary for POPC, which did not bind MG detectably (Supplementary Fig. 9). The normalized fluorescence ($F$) was plotted for binding curve analysis and fitted in Origin Pro 9 by the Hill equation[77] $F = (F_{max} \times C^n)/(K_D^n + C^n)$, where $C$ is the concentration of ligand, $n$ is the Hill coefficient, and $K_D$ is the dissociation constant. Since the interaction is known to be 1:1, curve fitting to the Hill equation was implemented with $n = 1$. The reported $K_D$ is the average of at least three independent experiments. During $K_D$ measurement, the average concentration of RNA was estimated to be ~58 nM (for RNA inside vesicles after column purification, estimated using RT-qPCR, method described below) or 0.25 μM (for RNA outside the vesicles). The concentrations of RNA were well below the $K_D$, which allowed the use of Hill equation instead of the analytical quadratic solution[78]. To confirm the use of the Hill equation, $K_D$s calculated by fitting to the quadratic solution (Supplementary Table 1) were found to be within experimental error of $K_D$s calculated by the Hill equation (for encapsulated samples, an RNA concentration of 58 nM was used for the fit to the quadratic solution). $K_D$ measurements in the presence of crowding agents were performed in analogous fashion without vesicles.

**Measurement of encapsulated Mg$^{2+}$ concentration.** The internal Mg$^{2+}$ concentration of the vesicles was measured using mag-fura-2 according to the procedures described earlier[39]. Since the mag-fura-2 assay was found to be insensitive to Mg-citrate concentration, these experiments were performed using 1 mM MgCl$_2$ instead of Mg-citrate. Although 1 mM MgCl$_2$ affects the stability of MA vesicles, experiments were conducted before substantial aggregation was detected by DLS.

**Melting transitions of RNA measured by fluorescence.** The fluorescence of the aptamers was measured at different temperatures using a Fluoromax 4C (Horiba) with a Peltier-controlled temperature attachment (Model F-3004, Horiba). The fluorescence of the aptamer–dye complex was recorded in 3 °C increments from 1 to 70 °C, with a 10 min incubation at each interval. Fluorescence intensity spectra (excitation at 617 nm, emission at 650–750 nm) were recorded and the baseline was subtracted from raw fluorescence values. Melting curves were fitted in Origin Pro 9 software using the Boltzmann sigmoidal equation $F = F_{min} + (F_{max} - F_{min})/(1 + \exp((T_t - T)/s))$, where $F$ refers to fluorescence ($\lambda = 660$ nm) and $F_{min}$ and $F_{max}$ are the minimum and maximum fluorescence, respectively, $T$ is temperature, $T_t$ is the transition temperature, and $s$ is a fitting parameter. $T_t$ values presented here are average of three independent experiments. Fatty acid vesicles are known to be thermostable across this temperature regime[79]. For samples containing vesicles, the background fluorescence ($\lambda = 660$ nm) of a control sample containing dye in the presence of vesicles (without aptamer) was subtracted from the fluorescence of vesicles with aptamer-bound dye at each temperature.

**Melting curve by CD spectroscopy.** Temperature-dependent CD spectra were acquired using a JASCO J-1500 spectrophotometer (JASCO International Co. Ltd, Tokyo, Japan) equipped with a Peltier-controlled cell holder (model PTC-517, JASCO). A sample containing 3.8 μM RNA, 12 μM MG, 1.7 mM Tris, 10 mM HEPES, 1 mM Mg-Citrate, 100 mM KCl, 0.2 M bicine, pH 8.5 was prepared. The CD signal of the aptamer was monitored at 264 nm in 5 °C increments from 1 to 91 °C, with a 10 min incubation at each interval. The spectra were recorded using a 1 mm path length cuvette. Melting curves were fitted in Origin Pro 9 software using the Boltzmann sigmoidal equation $\theta = \theta_{min} + (\theta_{max} - \theta_{min})/(1 + \exp((T_m - T)/s))$, where $\theta$ refers to the ellipticity at 264 nm and $\theta_{min}$ and $\theta_{max}$ are the minimum and maximum $\theta$, respectively, $T$ is temperature, $T_m$ is the melting temperature, and $s$ is a fitting parameter.

**SHAPE assay.** The malachite green aptamer was adapted for the SHAPE assay by adding a T7 promoter, 5′ and 3′ linker regions, and a 3′ reverse transcription (RT) primer binding site[68] (Fig. 5b, template DNA sequence (aptamer region in bold text): TAATACGACTCACTA-TAGGGCCTTCGGGCCAA**GGATCCCGACTGGCGA-GAGCCAGGTAACGAATGGATCC**TCGATCCGGTTCGCCGGATC-CAAATCGGGCTTCGGTCCGGTTC). The ssDNA was obtained from IDT and amplified through 30 cycles of PCR using KAPA Taq ReadyMix PCR kit (Kapa Biosystems) in Bio-Rad C1000 thermal cycler (forward primer sequence: 5′-TAAT

ACGACTCACTATAGGGCCTTCGG-3′; reverse primer sequence: 5′-GAACCGG ACCGAAGCCCG-3′). PCR was carried out in 50 μL reactions (14 μM DNA template, 0.5 μM each primer, and 1× KAPA Taq Ready Mix), with initial melting at 95 °C for 3 min, followed by 30 cycles of 95 °C for 30 s, 57 °C for 30 s, 72 °C for 1 min, with an additional final extension at 72 °C for 1 min. The dsDNA transcription template was purified by 8% native-PAGE using gel extraction with the G-Biosciences electroelution kit and quantified by spectrophotometry (yielding 50–250 ng μL$^{-1}$ in 1× TAE buffer). RNA was transcribed in 30 μL reactions (containing 2 μL of DNA template, 10 μL NTP buffer mix, and 2 μL T7 RNA polymerase) using the HiScribe T7 Quick High Yield RNA Synthesis Kit (NEB) with a 16 h incubation time at 37 °C. Transcripts were 8% native-PAGE purified, extracted by electroelution into 1× TAE, quantified with spectrophotometry, and pooled for a stock solution. RNA for all treatments was annealed simultaneously using the method described above. Encapsulation and purification of vesicles containing the MG aptamer SHAPE construct were achieved using the methods described above. Varying concentrations of MG dye were added, keeping the same RNA concentration in each MG concentration series. One hundred and thirty millimolar NMIA stock was prepared in anhydrous DMSO and diluted to a final reaction concentration of 12 mM in the samples containing RNA (0.2 μM for RNA outside vesicles and ~50 nM for RNA inside vesicles). After a 10 h incubation of the RNA sample with MG at room temperature, NMIA was added as the acetylation agent for the SHAPE reaction. Negative control samples received the same volume of anhydrous DMSO. All samples were incubated at 10 °C overnight to allow diffusion and reaction of NMIA (the low temperature extends the half-life of NMIA[68]). Vesicle rupturing was achieved with 0.6% Triton X-100 and a 30 min, room temperature incubation. NMIA-modified RNA was recovered by ethanol precipitation with 0.3 M sodium acetate and 15 μg of GlycoBlue, followed by a wash with ice-cold 95% ethanol. Pellets were dried and resuspended in 0.5× TE buffer. SuperScript IV reverse transcriptase (Invitrogen) was used for cDNA synthesis and for transcript sequencing reactions, which used a 2:1 ratio of dNTPs to ddNTPs. Both the reverse transcription reactions and the sequencing reactions used a 5′ fluorescently tagged (Rhodamine Green) RT primer to label products for visualization on a gel (RT Primer sequence: /5RhoG-XN/GAA CCG GAC CGA AGC CCG). Reverse transcription reactions were conducted as follows: 10 μL of NMIA-reacted RNA (or 5 μL of transcript for sequencing reactions), 2 μL of 2 μM reverse primer, and 1 μL 10 μM dNTP mix were mixed and incubated at 65 °C for 5 min and annealed on ice for 5 min. Four microliters Superscript IV buffer mix, 1 μL 0.1 M DTT, and 1 μL Superscript IV reverse transcriptase were added and the reaction was incubated at 55 °C for 10 min, and then deactivated at 80 °C for 10 min. RNA was removed by base hydrolysis by addition of 1 μL of 4 M NaOH followed by neutralization by 4 μL of 1 M Tris-HCl. Samples were analyzed by 12% urea-PAGE run on an Apogee S2 sequencing gel apparatus at 1200 V for 6 h. The gel was imaged using an Amersham Typhoon 5 or Typhoon FLA9500 from GE, and band quantitation was performed using GE ImageQuant software. Band intensities were adjusted for gel background intensity, and ratiometric measurements were made to account for variations in loading and/or reverse transcription efficiency. The gel band intensity ratios were baseline-subtracted and then normalized to 1 (by dividing the maximum of the intensity ratios) for fitting to the Boltzmann equation. $K_D$ values presented here are the average of three independent experiments. Experiments were performed in triplicate.

**Quantification of MG aptamer.** The amount of encapsulated RNA present in a sample of vesicles was determined by RT-qPCR. Although primers could not be designed for the MG aptamer itself due to its small size, the MG aptamer SHAPE construct was amenable to RT-qPCR. The MG construct used for SHAPE was encapsulated in vesicles as described above, and the purified vesicles were ruptured with 0.6% Triton X-100 incubated for 30 min at room temperature. RNA stocks were serially diluted two-fold in 4 mM MA containing 10 mM HEPES, 1 mM Mg-Citrate, 100 mM KCl, 0.2 mM bicine, pH 8.5, and 0.6% Triton X-100 to create a standard curve ranging from 0.08 to 0.002 μM (Supplementary Fig. 12). One microliter of the RNA stock was added to 19 μL of Bio-Rad iTaq Universal SYBR Green One-Step RT-qPCR reagent to set up 20 μL reactions following the manufacturer's protocol. Primers were obtained from IDT (forward primer sequence: 5′-GGC CTT CGG GCC AAG GAT C-3′; reverse primer sequence: 5′-GAA CCG GAC CGA AGC CCG-3′) to produce a 95 bp amplicon. Reverse transcription was performed for 10 min at 50 °C, followed by 40 cycles of PCR (melting for 10 s at 95 °C; annealing and extension for 30 s at 60 °C) and melting curve analysis (65–95 °C in 0.5 °C increments). RT-qPCR was performed on a Bio-Rad C1000 thermal cycler with a Bio-Rad CFX96 Real-Time PCR block.

**Data availability.** Data generated during this study are included in this published article (and its Supplementary Information file). Additional data (e.g., DLS files) are available from the corresponding author on reasonable request.

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

## Acknowledgements

We thank Omar Saleh, Kate Adamala, Jack Szostak, Jerry Joyce, and Ulrich Muller for advice, and Andrej Luptak for advice and materials for the SHAPE assay. We thank the Biological Nanostructures Laboratory at the California NanoSystems Institute at UCSB for use of equipment. Funding was provided by the Simons Foundation (grants 290356 and 481325), NASA (NNX16AJ32G), the Searle Scholars Program, and the Institute for Collaborative Biotechnologies (W911NF-09-0001 from the U.S. Army Research Office).

## Author contributions

R.S. and S.V. conducted the experiments. R.S. and I.A.C. conceived the experiments and wrote the paper.

## Additional information

**Competing interests:** The authors declare no competing interests.

