## [Peer Review File · Nature Communications]

Reviewers' comments:

Reviewer #1 (Remarks to the Author):

In this manuscript, Chen and coworkers investigated that the stability of RNA aptamer and interactions the RNA aptamer in the inside fatty acid vesicles. As results, the binding affinity between the RNA aptamer and ligand was increased. Moreover, the RNA aptamer with ligand in the inside vesicles was stabilized although the RNA aptamer with ligand without ligand was destabilized. Interestingly, in the outside vesicles, the binding affinity between the RNA aptamer and ligand were stabilized and destabilized depending on the cosolutes. The authors concluded that the binding affinity of vesicle effectively 'chaperones' the RNA, consistent with an excluded volume mechanism due to confinement. It is new and very interesting phenomenon of RNA aptamer behavior in the inside and outside vesicles in this manuscript. The results will be important information for understand RNA behavior in cells from biological and medical fields. However, there are many points still unclear. This referee feels the authors should describe several points in detail as shown below and too preliminary to publish for this high level journal.

1. Overall, the authors should explain experimental procedure in more detail to show the adequacy of experiments. For example, the author should show explicitly how the authors confirmed the RNA location in or outside vesicles and how many RNAs present in vesicles.
2. There are several reports on nucleic acids inside vesicles or reverse micells (ex. J Am Chem Soc. 2009 1313806-7). The authors should compare their results and results in the present manuscript.
3. The authors did not carry out the experiments to understand mechanism of RNA behavior difference inside and outside vesicles.
4. The authors concluded that the excluded volume due to confinement was important to determine the RNA behavior. If so, the author should investigated the effect of vesicle size systematically and the number of RNA molecules to the binding affinity and RNA stability. Moreover, the RNA behavior using different size of aptamer in vesicles will be informative.
5. The authors found very interesting phenomenon in the outside vesicles. The binding affinity between the RNA aptamer and ligand were stabilized and destabilized depending on the cosolutes independent of cosolute size. These results indicated that the excluded volume was not so important.

Reviewer #2 (Remarks to the Author):

This emergence of functional RNAs and their replication and encapsulation inside primitive compartments is one of the key questions in the origin of life field. The manuscript „Lipid Vesicles Chaperone Encapsulated RNA“ by Saha et al. describes how encapsulation of the malachite green aptamer into small lipid vesicles leads to a chaperone effect that assists folding of this RNA into its active conformation. The authors measure the increased folding properties of the aptamer using ligand titration, melting curves, and SHAPE analysis. They conclude that the vesicle exerts an excluded volume mechanism, which leads to an increase in RNA fitness. The authors discuss that the encapsulation of RNAs in vesicle formed from simple amphiphiles could have increased the chance for the emergence of functional sequences during abiogenesis. A chaperone effect of vesicles for RNA folding has not been reported previously.

The manuscript is organized, well written, easy to follow, and cited appropriately.

However, there are some major points that require clarification and further experimental evidence:

- The authors do not state anywhere in the manuscript which aptamer concentration was used to

determine MG-affinities inside and outside of the vesicles. However, this is of great importance for the interpretation of the data. Given the low K_d for MG binding, use of the Hill-equation becomes inappropriate for fitting the binding curves even at low micromolar aptamers concentration and requires the analytic quadratic solution for 1:1 binding instead (e.g. see equation 8 in <https://dx.doi.org/10.1091%2Fmbc.E10-08-0683>). The resulting K_d s from a new fit might deviate significantly from the ones obtained from fitting the Hill-equation and may thus require re-interpretation. The referee is aware that a direct measurement of the aptamers concentration after encapsulation in vesicles is difficult. However, an approximation the encapsulated amount of aptamers should be possible e.g. based on the chromatogram from the SEC-based purification or qRT-PCR based methods.

- The authors measure thermal denaturation of the aptamers indirectly using malachite green fluorescence, thus assuming that the change in fluorescence reports directly on global unfolding of the aptamer. However, the temperature has also an effect on the MG-aptamer binding equilibrium independent from aptamers unfolding. Supporting Figure 6 (melting curve in solution without fatty acids) would therefore greatly benefit from a validation of this assumption by directly measuring the unfolding of the aptamers using CD-spectroscopy.
- The destabilization of the aptamers in presence of MA seems quite drastic (34.2°C vs 4.8°C). While encapsulation seems to partially rescue aptamers folding, the authors should comment more on a possible mechanism that leads to this strong detrimental effect of MA on RNA folding. Has this also been observed for other systems?
- Can the authors comment on the possibility of binding interactions between MG and vesicle membrane in the absence of aptamer? Would this change the effective concentration inside the vesicle and lead to an apparent shift in K_d ? It should be straightforward to check if MG is accumulating, e.g. by equilibrating with MG, size exclusion then UV-Vis.
- Similar to the titrations based on MG-fluorescence, there is no clear information about the concentrations of aptamer used for the SHAPE analysis. However, the importance of this information for the relative amount of aptamers-MG complex applies also here. Similarly, it would be helpful to include the gel of the SHAPE analysis of aptamers inside of the vesicles in the Supplementary Material.

Minor point:

- On page 6 line 4 the authors state: "Unencapsulated MG aptamer outside 'empty' vesicles had a lower K_D ($6.4 \pm 1.4 \mu\text{M}$), indicating that the presence of lipid reduced the affinity of the aptamers (Fig. 2a and Supplementary Fig. 3b)." It should be "...vesicles had a higher K_D ..."

Reviewer #3 (Remarks to the Author):

The manuscript by Saha et. al describes how encapsulation inside lipid vesicles affects properties of functional RNA. Authors postulate that the encapsulation in liposomes increases stability of the RNA fold, providing a potential mechanism for the interaction of primitive membranes with RNA during the prebiotic evolution and the origin of life processes.

This seems like an interesting idea, but authors did not investigate this idea thoroughly. Without more complete experimental data, it is hard to evaluate the general claims Authors make about chaperoning effects of liposomes on RNA.

The paper, while meritorious and containing conclusions worthy of publication in Nature Communications, is quite short, and the addition of several additional experiments would greatly improve the quality of the manuscript and support Author's conclusions.

Major concerns:

- while the Authors claim that this is a general effect, the experiments use very limited set of conditions. Only citrate was used as a chelating agent, and experiments were done with one kind of RNA aptamer in limited number of liposome types. To claim true prebiotic significance of this work, it would be very valuable to see data for other kinds of fatty acid – fatty alcohol – glycerol ester compositions of lipids, commonly used in prebiotic chemistry literature.

- Authors aim to investigate is “confinement inside vesicles could improve ribozyme activity and folding”, yet there is no work in this paper directly investigating any actual ribozyme. Authors cite previous work on hammerhead ribozyme in liposomes, but they also mention work showing that presence of liposomes is detrimental to hammerhead activity (references 39 and 55). Other than citing prior work, the Authors make no effort to test their liposome chaperoning effect on any active ribozyme, like hammerhead or other catalytic RNA.

- Authors hypothesize that crowding phenomena can explain the chaperoning effect of liposomes on RNA. It would be very valuable to see experiments done with liposomes of many different sizes, as the crowding effect in the lumen will be vastly different in 100nm vesicles vs 1 or 2um ones.

Minor point:

- Figure 5 is purely speculative, it's not real data. It does not bring anything new to the narrative. If it's meant to be a concept figure, then figure 1 is redundant.

- a lot of the paper, and most of the SI information, is focused on characterizing previously well-characterized systems, particularly on detailed characterization of MG aptamer. It is unclear what is new in this work, or why was this needed.

- when estimating magnesium level in presence of fatty acid liposomes, was the chelating effect of lipids included in calculating the available free magnesium?

- what kind of lipids were used to make vesicles on Figure 3?

- when measuring K_d of ligand in presence of liposomes, it would be important to control for the affinity of the ligand to the bilayer membrane.

Malachite green is hydrophobic and it would be easy to see how it binds to, or embeds, into hydrophobic lipid bilayer, which would shift the apparent measured K_d 's in presence of liposomes. I would suggest malachite green leakage assay, or co-purification (mix with empty liposomes, purify to see if malachite green elutes with liposome fraction).

Reviewer #1 (Remarks to the Author):

In this manuscript, Chen and coworkers investigated that the stability of RNA aptamer and interactions the RNA aptamer in the inside fatty acid vesicles. As results, the binding affinity between the RNA aptamer and ligand was increased. Moreover, the RNA aptamer with ligand in the inside vesicles was stabilized although the RNA aptamer with ligand without ligand was destabilized. Interestingly, in the outside vesicles, the binding affinity between the RNA aptamer and ligand were stabilized and destabilized depending on the cosolutes. The authors concluded that the binding affinity of vesicle effectively 'chaperones' the RNA, consistent with an excluded volume mechanism due to confinement. It is new and very interesting phenomenon of RNA aptamer behavior in the inside and outside vesicles in this manuscript. The results will be important information for understand RNA behavior in cells from biological and medical fields. However, there are many points still unclear. This referee feels the authors should describe several points in detail as shown below and too preliminary to publish for this high level journal.

1. Overall, the authors should explain experimental procedure in more detail to show the adequacy of experiments. For example, the author should show explicitly how the authors confirmed the RNA location in or outside vesicles and how many RNAs present in vesicles.

Reply 1. We have now included more experimental details in the Methods, as follows. The new text is shown in bold. Methods that describe new experiments are bold and underlined.

- a) Preparation of fatty acid vesicles. Previous methodology was adopted in order to prepare vesicles from MA **using the pH-drop method⁴¹**. Briefly, **3.7 μ L of neat MA** was mixed with one equivalent of **1 M KOH (15 μ L)** in water to produce a clear solution of **800 mM MA (micellar)**. To this solution **120 μ L of 1 M bicine** and **60 μ L of 10X salt mix (100 mM HEPES, 10 mM Mg-Citrate, 1 M KCl, pH 8.5)** and water were added to obtain a visibly turbid solution of **25 mM MA in 0.2 M bicine, 10 mM HEPES, 1 mM Mg-Citrate, 100 mM KCl, pH 8.5**. **For mixed composition lipid vesicles, the neat oils were mixed before dispersion in buffer solutions as previously described³⁹**. The molar ratio of GMM and MAOH to MA was **1:2 and 1:10, respectively.** *(page 14)*

- b) **Determination of CAC of MA:MAOH vesicles. The absorbance at 600 nm was monitored with serial dilution using the cuvette-based application of the IMPLEN P300 nanophotometer. 2 ml of a 4.3 mM vesicle suspension was prepared and the solution was serially diluted with 100-600 μ l of buffer solution containing 10 mM HEPES, 1 mM Mg-Citrate, 100 mM KCl, 0.2 M bicine, pH 8.5. The point of inflection of the curve was used to estimate the CAC of the vesicles (Supplementary Fig. 11).**
(Page 15)
- c) **Preparation of phospholipid vesicles.** To prepare phospholipid vesicles, 12 mM DOPG or 20 mM POPC dissolved in chloroform was dried by rotary evaporation onto a round-bottom flask and resuspended in 0.2 M bicine, 1 mM Mg-Citrate, 100 mM KCl, 10 mM HEPES, pH 8.5. **For preparing vesicles encapsulating RNA, the resuspension buffer also included annealed RNA (~40 μ M final concentration; see below for annealing procedure)...**
(Page 15)
- d) **Encapsulation of RNA.** The RNA aptamer solution (~57- 88 μ M in 50-150 μ L of 10 mM Tris-Cl, pH 8.5) was heated ... The annealed RNA was mixed with buffer so as to obtain a final concentration of ~33-58 μ M RNA in 0.6-2.5 mM Tris, 10 mM HEPES, 1 mM Mg-Citrate, 100 mM KCl, 0.2 M bicine, pH 8.5, after addition of MA. This solution was added to the MA micellar stock (see above) ... Vesicles were extruded (see above) and purified from unencapsulated (free) RNAs using a Sepharose 4B size exclusion column^{12,34} ... **For the purification of mixed composition vesicles, the corresponding lipid concentration used during purification was 1 mM for MA:GMM vesicles¹² and 2.8 mM for MA:MAOH vesicles (the CAC of MA:MAOH vesicles was estimated to be ~1.5 to 2 mM; Supplementary Fig. 11). For purification of phospholipid vesicles, the mobile phase did not contain additional lipids since the CAC of phospholipids is much lower (<10⁻⁸ M⁸⁰) than the lipid concentrations used here.**
(Page 16)
- e) **Preparation of empty vesicles exposed to RNA.** A 50 μ L solution was prepared containing 4 μ M RNA aptamer, 6.6 mM Tris, 10 mM HEPES, 1 mM Mg-Citrate, 100 mM KCl, 0.2 M bicine, pH 8.5. This solution was added to 100-150 μ L of preformed MA vesicles (25 mM MA, 10 mM HEPES, 1 mM Mg-Citrate, 100 mM KCl, 0.2 M bicine, pH 8.5) and 200-250 μ L of buffer (4 mM MA micelles, 10 mM HEPES, 1 mM Mg-Citrate, 100 mM KCl, 0.2 M bicine, pH 8.5) to obtain a solution containing 0.5 μ M RNA (non-encapsulated) and 8.8-11 mM MA. For K_D measurements, 20 μ L of this solution was added to 20 μ L of solution containing 4 mM MA, 10 mM HEPES, 1 mM Mg-Citrate, 100 mM KCl, 0.2 M bicine, pH 8.5, and varying concentrations of MG, and treated as described below. For MA:GMM and MA:MAOH vesicles, the same procedure was followed except that the buffer included the appropriate concentration (\geq CAC) of the corresponding lipid solution. For phospholipid vesicles, no additional lipid was added during dilution steps due to the low CAC of these lipids.
(Page 16)
- f) **Dissociation constant (K_D) measurement.** MG was dissolved in 4 mM MA solution containing 10 mM HEPES, 1 mM Mg-Citrate, 100 mM KCl, 0.2 mM bicine, pH 8.5, to obtain a stock solution of 1-80 μ M MG. ... The MG dye was excited at 617 nm and the emission monitored at 660 nm. Fluorescence intensity was normalized to a minimum of 0 and a maximum of 1. In cases including negatively charged membranes, to account for fluorescence signal arising from MG bound to the membrane (Supplementary Fig. 2b), the background fluorescence ($\lambda = 660$ nm) of a corresponding control sample containing the dye in the presence of the same concentration of vesicles (without aptamer) was subtracted from the fluorescence ($\lambda = 660$ nm) of vesicles with aptamer-bound dye (encapsulated or non-encapsulated). **This background correction was not necessary for POPC, which did not bind MG detectably (Supplementary Fig. 9). The normalized fluorescence (F) was plotted ... During K_D measurement, the average concentration of RNA was estimated to be ~58 nM (for RNA inside vesicles after column purification, estimated using RT-qPCR, method described below) or 0.25 μ M (for RNA outside the vesicles). The concentrations of RNA were well below the K_D , which allowed the use of Hill equation instead of the analytical quadratic solution⁸². To confirm the use of the Hill equation, $K_{D,S}$ calculated by fitting to the quadratic solution (Supplementary Table S1) were found to be within experimental error of $K_{D,S}$ calculated by the Hill equation (for encapsulated samples, an RNA concentration of 58 nM was used for the fit to the quadratic solution)...**
(page 17)

- g) **Melting transitions of RNA measured by fluorescence. ... Fluorescence intensity spectra (excitation at 617 nm, emission at 650 - 750 nm) were recorded and the baseline was subtracted from raw fluorescence values... For samples containing vesicles, the background fluorescence ($\lambda=660$ nm) of a control sample ...** (Page 18)
- h) **Melting curve by circular dichroism (CD) spectroscopy. Temperature-dependent CD spectra were acquired using a JASCO J-1500 spectrophotometer (JASCO International Co. Ltd, Tokyo, Japan) equipped with a Peltier-controlled cell holder (model PTC-517, JASCO). A sample containing 3.8 μ M RNA, 12 μ M MG, 1.7 mM Tris, 10 mM HEPES, 1 mM Mg-Citrate, 100 mM KCl, 0.2 M bicine, pH 8.5 was prepared. The CD signal of the aptamer was monitored at 264 nm in 5°C increments from 1°C to 91°C, with a 10 min incubation at each interval. The spectra were recorded using a 1 mm path length cuvette. Melting curves were fitted in Origin Pro 9 software using the Boltzmann sigmoidal equation $\theta = \theta_{min} + (\theta_{max} - \theta_{min}) / (1 + \exp((T_m - T)/s))$, where θ refers to the ellipticity at 264 nm and θ_{min} and θ_{max} are the minimum and maximum θ , respectively, T is temperature, T_m is the melting temperature, and s is a fitting parameter.** (Page 18)
- i) **SHAPE assay... PCR was carried out in 50 μ L reactions (14 μ M DNA template, 0.5 μ M each primer, and 1X KAPA Taq Ready Mix), with initial melting at 95 °C for 3 min, followed by 30 cycles of 95 °C for 30 s, 57 °C for 30 s, 72 °C for 1 min, with an additional final extension at 72 °C for 1 min. ... (yielding 50-250 ng/ μ L in 1X TAE buffer). RNA was transcribed in 30 μ L reactions (containing 2 μ L of DNA template, 10 μ L NTP buffer mix, and 2 μ L T7 RNA polymerase) ... final reaction concentration of 12 mM in the samples containing RNA (0.2 μ M for RNA outside vesicles and ~50 nM for RNA inside vesicles) ... Reverse transcription reactions were conducted as follows: 10 μ L of NMIA-reacted RNA (or 5 μ L of transcript for sequencing reactions), 2 μ l of 2 μ M reverse primer, and 1 μ l 10 μ M dNTP mix were mixed and incubated at 65°C for 5 minutes and annealed on ice for 5 minutes. 4 μ l Superscript IV buffer mix, 1 μ l 0.1 M DTT, and 1 μ L Superscript IV reverse transcriptase were added and the reaction was incubated at 55 °C for 10 minutes, and then deactivated at 80 °C for 10 minutes...** (Page 18)
- j) **Quantification of MG aptamer. The amount of encapsulated RNA present in a sample of vesicles was determined by RT-qPCR. Although primers could not be designed for the MG aptamer itself due to its small size, the MG aptamer SHAPE construct was amenable to RT-qPCR. The MG construct used for SHAPE was encapsulated in vesicles as described above, and the purified vesicles were ruptured with 0.6% Triton X-100 incubated for 30 minutes at room temperature. RNA stocks were serially diluted 2-fold in 4 mM MA containing 10 mM HEPES, 1 mM Mg-Citrate, 100 mM KCl, 0.2 mM bicine, pH 8.5, and 0.6% Triton X-100 to create a standard curve ranging from 0.08 μ M - 0.002 μ M (Supplementary Fig. 12). 1 μ l of the RNA stock was added to 19 μ L of Bio-Rad iTaq Universal SYBR Green One-Step RT-qPCR reagent to set up 20 μ l reactions following the manufacturer's protocol. Primers were obtained from IDT (Forward Primer sequence: 5'- GGC CTT CGG GCC AAG GAT C -3'; Reverse Primer sequence: 5'- GAA CCG GAC CGA AGC CCG -3') to produce a 95 bp amplicon. Reverse transcription was performed for 10 minutes at 50 °C, followed by 40 cycles of PCR (melting for 10 s at 95 °C; annealing and extension for 30 s at 60 °C) and melting curve analysis (65-95 °C in 0.5 °C increments). RT-qPCR was performed on a Bio-Rad C1000 thermal cycler with a Bio-Rad CFX96 Real-Time PCR block.** (Page 20)

To confirm the location of RNA, we now include a chromatogram of the separation of vesicles encapsulating RNA from free (unencapsulated) RNA, as new **Supplementary Figure 1**. To follow literature precedent, we repeated our experiments with encapsulated aptamer using separation with Sepharose 4B resin (our previous separation protocol used Toyopearl), as

Sepharose 4B is used in established protocols (Chen, Roberts, and Szostak, Science 2004, 305:1474-1476; Hanczyc et al., Science 2003, 302:618-22). We include these references in the Methods section "Encapsulation of RNA". The results obtained using this resin are similar to the results we obtained previously.

To determine how many RNAs are present in vesicles, we performed quantitative RT-PCR on the vesicles encapsulating RNA. The Methods were added as given above (**Reply 1j**, "Quantification of MG aptamer"). We detail the results in a new paragraph of the Results, as below ("Vesicles containing myristoleic acid (MA) and encapsulation of RNA").

- k)* To determine the quantity of malachite green (MG) RNA encapsulated, we measured the amount of RNA encapsulated using RT-qPCR and compared it to the value expected from random encapsulation. In a solution containing 2 mM MA in vesicles (i.e., 6 mM total MA given a critical aggregation concentration (CAC) of 4 mM⁴¹), given a surface area per MA molecule of $\sim 68 \times 10^{-20} \text{ m}^2$ ⁴³, the total bilayer surface area in solution is $\sim 410 \text{ m}^2/\text{L}$. Since a single vesicle of diameter 60 nm has a surface area of $\sim 10^{-14} \text{ m}^2$, we estimate a concentration of $\sim 4.1 \times 10^{16}$ vesicles/L. The concentration of encapsulated RNA (after dispersion into bulk solution) was found to be on the order of $\sim 58 \text{ nM}$ by RT-qPCR, or 3.5×10^{16} molecules/L, so on average we estimate ~ 0.85 RNA molecules per vesicle. Given the volume of a single vesicle ($\sim 10^{-22} \text{ m}^3 \approx 10^{-19} \text{ L}$), the bulk-equivalent concentration of RNA in the encapsulated volume would be $\sim 14 \text{ }\mu\text{M}$. This value is roughly consistent with the expectation based on random enclosure of RNA during vesicle formation (see Methods). The discrepancy may be due to estimation inaccuracies (e.g., of vesicle diameter, determined by DLS in this study, or of MA surface area, which had been determined in the context of a semicrystalline phospholipid) or from losses caused by charge-charge repulsion between the RNA and MA surface. (Page 5)

2. There are several reports on nucleic acids inside vesicles or reverse micelles (ex. J Am Chem Soc. 2009 1313806-7). The authors should compare their results and results in the present manuscript.

Reply 2. We now discuss the paper mentioned by the reviewer, as well as a related study by Sarkar and Pal (Biopolymers 2006, 83, 675), in new text of the Discussion, as follows.

- a)* These results are consistent with prior work, in which small RNAs were probed by NMR spectroscopy and their structures appeared to be stabilized in reverse micelles⁷⁰, likely due to restricted local motion. In our work, we show that confinement (in vesicles) also leads to functional improvement for the MG aptamer. Previous studies of a small DNA duplex indicated that binding of ethidium bromine was decreased by DNA condensation in reverse micelles⁷¹. Thus while nonspecific intercalation in DNA may be inhibited by confinement, the specific binding interaction between an RNA aptamer and its target can be enhanced. (Page 13)

3. The authors did not carry out the experiments to understand mechanism of RNA behavior difference inside and outside vesicles.

Reply 3. Some experiments included in the manuscript were intended to address the mechanism of this effect, including altering vesicle charge and composition, measuring Mg^{2+} activity inside the vesicles, SHAPE experiments, melting transitions of RNA measured by fluorescence, control experiments outside vesicles, experiments with crowding agents, and also the new experiments with CD spectroscopy and measurements at high aptamer concentration. While SHAPE and

fluorescence-based experiments are informative, deeper structural characterization (e.g., by NMR) is not practical due to the low effective concentration of aptamer after encapsulation. We have expanded the Discussion regarding the mechanistic interpretation of the experiments (new text in bold) as well as Results describing the new experiments. See **Reply 7** regarding CD experiments.

- a) In general, several mechanisms may be considered ... First, encapsulation might increase the local concentration of the ligand ... However, this mechanism does not apply here, because the small molecule ligand equilibrates across the membrane **within few hours (Supplementary Fig. 4). The encapsulation of a charged polymer does perturb the equilibrium concentration of permeable ions of the opposite charge through the Gibbs-Donnan effect⁶¹. Qualitatively, this effect would increase the concentration of MG inside the vesicles. However, the size of this effect is very small, corresponding to an [MG] increase of 1.0017-fold inside the vesicles (assuming one RNA molecule encapsulated), well within experimental error. Thus an increased concentration of MG inside the vesicles is unlikely to explain the effect on affinity that was observed. Second, RNA-RNA interactions might be stabilizing and be enhanced inside a minority of the vesicle population that encapsulates multiple RNAs. Chance encapsulation of two RNAs in a vesicle of volume 10^{-19} L would correspond to ~ 30 μM RNA. However, the T_i of the MG aptamer was not increased when measured at this higher concentration. A related possibility would be that membrane-bound MG dye is more favorable for binding by the RNA compared to aqueous MG. However, the observation that confinement in POPC vesicles (which do not bind MG detectably, Supplementary Fig. 9) gives a similar effect as confinement in other membrane types indicates that membrane-bound MG is unlikely to cause the increased affinity. ... measurement of the internal $[\text{Mg}^{2+}]$ indicated no difference from the bulk concentration. This observation also supports the small size of the Gibbs-Donnan effect in this system. A caveat of the measurement of $[\text{Mg}^{2+}]$ is that the K_D of mag-fura-2 for Mg^{2+} is 1.5 mM^{62} , so the assay would not detect binding of Mg^{2+} by lipid through weaker interactions. (Page 10)**
- b) **These results indicate that the excluded volume effect is not necessarily the primary driver of the effect of crowding agents on the MG aptamer, and chemical interactions with cosolutes would be an important factor affecting aptamer affinity. However, unlike the excluded volume effect of crowding agents, the confinement effect of vesicles can be studied separately from the chemical interactions by experimental comparison with non-encapsulated RNA that is exposed to vesicles (RNA outside vesicles). This comparison indicates a similar increase in affinity observed for the five lipid compositions tested here. (Page 12)**
- c) **We found a large decrease in the T_i monitored by fluorescence when RNA was exposed to MA vesicles, indicating decreased stability of interactions relevant to ligand-RNA binding (e.g., contacts between ligand and RNA, or local tertiary contacts in the binding region). A substantial decrease in the transition temperature was also observed in the presence of 18% glucose (34°C vs. 18°C), indicating that this cosolute is also destabilizing. However, both encapsulation and the crowding effect of 18% dextran substantially increased the T_i (Fig. 4a and Fig. 6b), suggesting local structural stabilization. (Page 12)**
- d) **To determine whether aptamer-aptamer interactions could induce the stabilization observed with encapsulation, we measured the T_i of non-encapsulated aptamer at high concentration ($28 \mu\text{M}$) in the presence of empty MA vesicles. As expected, the T_i was not significantly altered at high concentration (Supplementary Fig 5b). (Page 7)**

4. The authors concluded that the excluded volume due to confinement was important to determine the RNA behavior. If so, the author should investigate the effect of vesicle size systematically and the number of RNA molecules to the binding affinity and RNA stability. Moreover, the RNA behavior using different size of aptamer in vesicles will be informative.

Reply 4. Vesicles extruded to 100 nm diameter have been used extensively in the literature due to the ability to produce a relatively monodisperse, unilamellar population. While a study of the

effect of vesicle size would be helpful, it is not feasible to prepare monodisperse, unilamellar fatty acid vesicles of varying size. Electroformation to produce giant unilamellar vesicles has not been successful with fatty acid vesicles (despite our prior attempts), and indeed is generally limited to neutral lipids. Hydration generally produces heterogeneous multilamellar vesicles, which must be extruded for greater uniformity. Extrusion through pores larger than 100 nm creates multilamellar vesicles. While these may be homogenized with respect to size through dialysis (e.g., Zhu and Szostak, PloS One 2009,4, e5009), the number of lamellae cannot be controlled to one. Multilamellar structures are problematic for interpretation, as one could not determine how aptamers encapsulated in the interior volume vs. aptamers sandwiched between lamellae would contribute to the outcome.

On the other hand, we attempted to produce smaller unilamellar vesicles (SUVs) by extrusion through smaller pores (30 nm). However, the vesicles produced are ~65 nm in hydrodynamic diameter, as assayed by DLS. This observation is consistent with knowledge that SUVs are unstable due to their high bending energy and fuse over time. In addition, SUVs would face challenges in separation of non-encapsulated RNA from vesicles.

Theoretical calculations on the effect of confinement size, based on the model of Zhou and Dill (cited in the manuscript), suggest that the stabilization energy would be only mildly dependent on confinement diameter above ~30 nm, although large effects are expected when the vesicle approaches the size of the RNA (new **Supplementary Figure 10**). Thus for the experimentally relevant size regime, the effect of confinement is not expected to be very sensitive to vesicle size. Similarly, it can be seen that the stabilization energy would be insensitive to typical variations in aptamer size (new **Supplementary Figure 10**). Thus experiments to find an effect of size variation are not practical given experimental constraints on vesicle preparation and typical aptamer sizes. We address this issue in the text and figure added to the Discussion.

- a) **In principle, varying the confinement size and the RNA size might allow further probing of this effect, although the quantitative variation of this effect within the regime of experimentally tractable vesicle sizes and aptamers of typical size is predicted to be small (Supplementary Fig. 10)^{45,75,76}.**
(Page 13)

The reviewer also raises the concern that RNA concentration may mediate the effects seen. It is possible that encapsulation of multiple RNAs in a small volume might enhance RNA-RNA interactions; if these interactions are stabilizing or enhance function, they could be responsible for the observed effect. While the concentration of RNA in the total encapsulation solution was 58 nM by quantitative RT-PCR, the 'concentration' inside the vesicle would be ~30 μ M if 2 molecules were encapsulated in one vesicle. We checked the transition temperature of the aptamer at this higher RNA concentration. The T_i was not significantly altered by increased RNA concentration. We include this experiment in the Results and Discussion (see **Reply 3a,d**; Supplementary Figure 5b).

5. The authors found very interesting phenomenon in the outside vesicles. The binding affinity between the RNA aptamer and ligand were stabilized and destabilized depending on the cosolutes independent of cosolute size. These results indicated that the excluded volume was not so important.

Reply 5. The reviewer notes that the RNA was affected by the identity of the cosolutes. Dextran, PEG and Ficoll all showed different effects. This suggests that, for crowding agents, the excluded volume effect may not be as significant as chemical interactions with the cosolutes (although the comparison between glucose and dextran suggests an excluded volume effect). However, for vesicles, the excluded volume effect can be separated from cosolute effects by the comparison between encapsulated RNA and RNA exposed to the same solution (but not encapsulated) (Figure 1). We now discuss this in the Discussion, as follows.

- a) These results indicate that the excluded volume effect is not necessarily the primary driver of the effect of crowding agents on the MG aptamer, and chemical interactions with cosolutes would be an important factor affecting aptamer affinity. However, unlike the excluded volume effect of crowding agents, the confinement effect of vesicles can be studied separately from the chemical interactions by experimental comparison with non-encapsulated RNA that is exposed to vesicles (RNA outside vesicles). This comparison indicates a similar increase in affinity observed for the five lipid compositions tested here. (page 12)

Reviewer #2 (Remarks to the Author):

This emergence of functional RNAs and their replication and encapsulation inside primitive compartments is one of the key questions in the origin of life field. The manuscript „Lipid Vesicles Chaperone Encapsulated RNA” by Saha et al. describes how encapsulation of the malachite green aptamer into small lipid vesicles leads to a chaperone effect that assists folding of this RNA into its active conformation. The authors measure the increased folding properties of the aptamer using ligand titration, melting curves, and SHAPE analysis. They conclude that the vesicle exerts an excluded volume mechanism, which leads to an increase in RNA fitness. The authors discuss that the encapsulation of RNAs in vesicle formed from simple amphiphiles could have increased the chance for the emergence of functional sequences during abiogenesis. A chaperone effect of vesicles for RNA folding has not been reported previously.

The manuscript is organized, well written, easy to follow, and cited appropriately.

However, there are some major points that require clarification and further experimental evidence:

The authors do not state anywhere in the manuscript which aptamer concentration was used to determine MG-affinities inside and outside of the vesicles. However, this is of great importance for the interpretation of the data. Given the low K_D for MG binding, use of the Hill-equation becomes inappropriate for fitting the binding curves even at low micromolar aptamers concentration and requires the analytic quadratic solution for 1:1 binding instead (e.g. see equation 8 in <https://dx.doi.org/10.1091%2Fmbc.E10-08-0683>). The resulting K_D s from a new fit might deviate significantly from the ones obtained from fitting the Hill-equation and may thus require re-interpretation. The referee is aware that a direct measurement of the aptamers concentration after encapsulation in vesicles is difficult. However, an approximation the encapsulated amount of aptamers should be possible e.g. based on the chromatogram from the SEC-based purification or qRT-PCR based methods.

Reply 6. We have now estimated the concentration of the encapsulated MG aptamer by RT-qPCR. For aptamer encapsulated in vesicles, the concentration in the total solution was found to be ~58 nM, which is well below the K_D . This was determined for the SHAPE construct of the

MG aptamer, which was encapsulated under similar conditions as the wild-type MG aptamer. The small size of the MG aptamer (38 nt) made primer design for PCR problematic, so the SHAPE construct of the aptamer was used. We now include this quantitation in the Results and Methods (see **Reply 1j, k**). For non-encapsulated aptamer, the RNA concentration was 0.2-0.25 μM . This is approximately one order of magnitude less than the K_D . To confirm that use of the Hill equation did not introduce inaccuracies, we fitted the data with both the Hill equation and the analytic quadratic equation. The K_{DS} determined from both fits are given in the new **Supplementary Table S1** and were found to be within experimental error of each other. We now describe this in the Methods (see **Reply 1f**).

The authors measure thermal denaturation of the aptamers indirectly using malachite green fluorescence, thus assuming that the change in fluorescence reports directly on global unfolding of the aptamer. However, the temperature has also an effect on the MG-aptamer binding equilibrium independent from aptamers unfolding. Supporting Figure 6 (melting curve in solution without fatty acids) would therefore greatly benefit from a validation of this assumption by directly measuring the unfolding of the aptamers using CD-spectroscopy.

Reply 7. The reviewer points out that the melting curve measured by fluorescence reports on disruption of the local structure of the aptamer-MG binding region, not on the global fold of the aptamer. CD spectroscopy reports on the melting of secondary structure in the RNA. Although it is not possible to obtain CD spectroscopy of the encapsulated RNA due to the low RNA concentration, we followed the melting curve by CD for non-encapsulated RNA in aqueous buffer, without vesicle (new **Figure 4b**), as suggested by the reviewer. The T_m of the aptamer measured by CD is 71 ± 2 $^{\circ}\text{C}$, which is substantially higher than the transition temperature measured by fluorescence (34.2 ± 2.3 $^{\circ}\text{C}$). This confirms that the melting studies should be interpreted more narrowly. In particular, we changed the interpretation of the fluorescence changes observed during melting studies as reporting on the local interactions of the ligand-binding site. We also change the wording from 'melting temperature (T_m)' to 'transition temperature (T_t)' to more accurately reflect this distinction. We note that measurement of the apparent T_m of a dye-binding aptamer by fluorescence has precedent in the literature, as it has been used to report on stability of the Spinach aptamer (Nat Methods 2013, 10, 1219). Changes to text are given below (new and modified text in bold). We thank the reviewer for this comment.

a) Results: Melting **transition** of the MG aptamer. To determine whether the RNA structure was stabilized by encapsulation, we **monitored the melting transition ... Measurement by fluorescence reports on interactions relevant to ligand binding and not necessarily on the overall fold of the RNA.** In the presence of empty MA vesicles (aptamers not encapsulated), **a single transition was observed with a transition temperature (T_t) of 4.8 ± 1.1 $^{\circ}\text{C}$... To understand whether the transition represented overall folding of the RNA or local interactions in the ligand-binding site, we measured the melting temperature (T_m) by CD spectroscopy. The T_m of the aptamer in aqueous buffer, without vesicles, monitored by CD, was found to be 71 ± 2 $^{\circ}\text{C}$ (Fig. 4b), indicating that T_t represents interactions in the ligand-binding site and not global folding or large transitions of secondary structure. (page 7)**

b) Discussion modified: see Reply 3c

c) Methods added: see Reply 1h

d) Introduction:

- changed "folding stability of the MG aptamer" to "local structural interactions of the MG aptamer"

- changed "characterize binding affinity and **RNA folding**" to "characterize binding affinity and **local RNA folding**"
- e) Abstract: changed "stabilizes the bound conformation" to "**locally** stabilizes the bound conformation"

The destabilization of the aptamers in presence of MA seems quite drastic (34.2°C vs 4.8°C). While encapsulation seems to partially rescue aptamers folding, the authors should comment more on a possible mechanism that leads to this strong detrimental effect of MA on RNA folding. Has this also been observed for other systems?

Reply 8. There is a precedent for negatively charged detergents inhibiting ribozyme activity, as well as several prior studies on CTAB, a positively charged detergent, which has been found to enhance activity of ribozymes. These studies support the interpretation that electrostatic effects are primarily responsible for the effect. Suga et al. also suggested a role for the hydrophobic effect in the interaction of lipids with RNA. We now summarize these studies in the related section of the Discussion. We also summarize the literature on stabilization due to crowding and confinement effects, as measured by melting temperature. Please also see **Reply 3c**.

- a) Interaction between RNA and lipids is believed to be mediated by both electrostatic interactions with the headgroup and hydrophobic interactions to nucleobases⁵³, and may be enhanced by an ordered bilayer⁵⁴.**Anionic detergents (e.g., sodium dodecyl sulfate) have been previously observed to completely inhibit a self-cleaving ribozyme, while non-ionic and zwitterionic detergents substantially enhanced activity⁵⁶. Conversely, the cationic detergent cetyltrimethylammonium bromide (CTAB) has been shown to promote self-cleaving ribozyme reactions^{57,58}, likely by increasing the rate of association and dissociation, potentially by 3-4 orders of magnitude⁵⁹. (page 9)**
- b) **This degree of stabilization is similar to previous observations and models. For example, encapsulation of α -lactalbumin in the pores of silica glass was found to increase the melting temperature by 32 °C⁶⁵. Using molecular dynamics simulations, an RNA pseudoknot was shown to be stabilized, relative to the extended hairpin structure, by crowding agents, with an expected melting temperature increase from 78 to 91 °C⁶⁶. A statistical-thermodynamic model of proteins in physiological crowding conditions predicts stabilization of folding transitions by 5-20 °C⁶⁷. It is also possible that the effect of confinement may be greater at certain local features (e.g., a pseudoknot or ligand-binding site) than for the global structure. (page 12)**

Can the authors comment on the possibility of binding interactions between MG and vesicle membrane in the absence of aptamer? Would this change the effective concentration inside the vesicle and lead to an apparent shift in K_D ? It should be straightforward to check if MG is accumulating, e.g. by equilibrating with MG, size exclusion then UV-Vis.

Reply 9. MG, a positively charged dye, does not appear to interact with POPC (new **Supplementary Figure 9**), as no detectable peak corresponding to vesicles is found when mixing POPC and MG. Light scattering from vesicles interferes with the UV-Vis signal. However, membrane-bound MG does fluoresce and can be used to detect MG during size exclusion chromatography. Confinement in POPC vesicles exhibits an increase in affinity similar to confinement in other vesicle types. This indicates that the effect on affinity does not only depend on MG binding to the membrane. However, MG does bind to MA membranes and this phenomenon contributes some background signal to these measurements, as shown in new **Supplementary Figure 2b**. This background was subtracted during K_D estimation as described in the Methods (see **Reply 1f**).

The chemical activity of MG should be the same inside and outside of the vesicles after equilibration. One perturbation to the equilibrium that can be considered is the Gibbs-Donnan effect, i.e., that a trapped polymer alters the equilibrium concentration of a positively charged species. This effect can be calculated and is quantitatively too small to account for an observable shift in K_D . Although some MG may be bound to the membrane at equilibrium depending on the membrane type, we are not aware of another physical mechanism that could create an accumulation of MG in the aqueous compartment inside vesicles compared to the external solution. A related possibility could be that the membrane-bound MG is more favorable for binding by the RNA compared to aqueous MG. However, the observation that confinement in POPC vesicles (which do not bind MG detectably) gives a similar effect to confinement in MA vesicles indicates that membrane-bound MG is unlikely to cause the increased affinity. We include these issues in the Discussion (see **Reply 3a**).

Similar to the titrations based on MG-fluorescence, there is no clear information about the concentrations of aptamer used for the SHAPE analysis. However, the importance of this information for the relative amount of aptamers-MG complex applies also here. Similarly, it would be helpful to include the gel of the SHAPE analysis of aptamers inside of the vesicles in the Supplementary Material.

Reply 10. The quantitative RT-PCR described above was performed on the SHAPE construct for the MG aptamer rather than the wild-type MG aptamer, due to technical problems with primer design for the short MG aptamer (38 nt). Please see **Reply 1i,j,k**.

A gel of the SHAPE assay of encapsulated aptamer was also added as new **Supplementary Figure 6**.

Minor point:

On page 6 line 4 the authors state: “Unencapsulated MG aptamer outside ‘empty’ vesicles had a lower K_D ($6.4 \pm 1.4 \mu\text{M}$), indicating that the presence of lipid reduced the affinity of the aptamers (Fig. 2a and Supplementary Fig. 3b).” It should be “...vesicles had a higher K_D ...”

Reply 11. This correction was made.

Reviewer #3 (Remarks to the Author):

The manuscript by Saha et. al describes how encapsulation inside lipid vesicles affects properties of functional RNA. Authors postulate that the encapsulation in liposomes increases stability of the RNA fold, providing a potential mechanism for the interaction of primitive membranes with RNA during the prebiotic evolution and the origin of life processes.

This seems like an interesting idea, but authors did not investigate this idea thoroughly. Without more complete experimental data, it is hard to evaluate the general claims Authors make about chaperoning effects of liposomes on RNA.

The paper, while meritorious and containing conclusions worthy of publication in Nature Communications, is quite short, and the addition of several additional experiments would greatly improve the quality of the manuscript and support Author's conclusions.

Major concerns:

- while the Authors claim that this is a general effect, the experiments use very limited set of conditions. Only citrate was used as a chelating agent, and experiments were done with one kind of RNA aptamer in limited number of liposome types. To claim true prebiotic significance of this work, it would be very valuable to see data for other kinds of fatty acid – fatty alcohol – glycerol ester compositions of lipids, commonly used in prebiotic chemistry literature.

Reply 12. We added studies of the MG aptamer encapsulated in MA:GMM (fatty acid with its glycerol monoester) and MA:MAOH (fatty acid with its fatty alcohol). The K_D shift in these vesicles is similar to that observed for the other vesicle types investigated (MA, POPC, DOPG). These figures are now added (new **Figure 2b,c**) and the related methods are included (see **Reply 1a,b,d,e**). We also include new text to report the findings in the Results.

- a) **To mimic more prebiotically plausible protocellular conditions, we repeated these experiments with membranes composed of a mixture of lipids known to increase the stability of fatty acid vesicles⁴⁶. A similar increase in affinity from confinement was found using vesicles composed of MAOH or GMM mixed with MA (Fig. 2b and 2c).** (page 7)

We studied the MG aptamer without magnesium citrate (now **Figure 6c**), and we observed the same increase in affinity when encapsulated. We modify the text of the results to clarify this, and we moved the figure from the supplement to the main body of the manuscript.

- b) **Also, a similar shift in K_D was observed upon encapsulation inside vesicles in the presence or absence of Mg^{2+} (Fig. 3a and 6c). This observation indicates that the effect of encapsulation may be observed in other salt conditions.** (page 8)

While the use of multiple aptamers would be an interesting extension of this work, I believe this would go beyond the scope of this study. Many studies of crowding agents or confinement on RNA activity have focused on a single RNA, usually a ribozyme (see Paudel and Rueda, JACS 2014, 136:16700; Kilburn et al., JACS 2010, 132:8690; Kilburn et al., JACS 2013, 135:10055; Desai et al., J. Biol. Chem 2014, 289:2972; Strulson et al., Biochem. 2013, 52:8187; Tyrrell et al., Biochem. 2015, 54:6447; Lee et al., Nuc. Acids Res. 2015, 43:1170; Strulson et al., Nat. Chem. 2012, 4:941, and others).

To address this issue, we have made several changes to the text to reduce the claim that this is a general effect. The title was changed to "Lipid vesicles chaperone encapsulated RNA" to "Lipid vesicles chaperone **an encapsulated RNA aptamer**". The word "aptamer" is replaced by "**malachite green aptamer**" in several instances in the Abstract, Introduction, and Discussion. Other text changes are given as follows.

- c) Discussion: "Further studies are needed to understand how different reactions would be affected." was changed to "Further studies are needed to understand how different **aptamers or** reactions would be affected." (page 10)
- d) The following sentence was added to the final paragraph: "**An important caveat is that the effect of**

encapsulation was only studied for one aptamer here, in a limited number of chemical conditions and vesicle compositions; further work would be needed to assess the generality of the effect."

(page 14)

- Authors aim to investigate is "confinement inside vesicles could improve ribozyme activity and folding", yet there is no work in this paper directly investigating any actual ribozyme. Authors cite previous work on hammerhead ribozyme in liposomes, but they also mention work showing that presence of liposomes is detrimental to hammerhead activity (references 39 and 55). Other than citing prior work, the Authors make no effort to test their liposome chaperoning effect on any active ribozyme, like hammerhead or other catalytic RNA.

Reply 13. This misunderstanding was our error in writing. This paper studies aptamers, not ribozymes. As mentioned previously, we changed the title to read "**RNA aptamer**" instead of "RNA". The text mentioned by the reviewer has been changed to "confinement inside vesicles could improve **RNA aptamer** activity."

The following additional changes were made to ensure that ribozymes are only mentioned when discussing prior work specifically regarding ribozymes.

- a) Introduction, 1st paragraph: "... encapsulation itself could improve ribozyme function... studying how encapsulation affects functional RNA is important" was changed to "... encapsulation itself could improve **RNA function ... studying how encapsulation affects functional RNA (e.g., RNA aptamers)**". *(page 3)*
- b) Introduction, 3rd paragraph: "emergence of ribozymes" to "emergence of **functional RNA**". *(page 4)*
- c) Discussion, clarification regarding the encapsulated hammerhead ribozyme: "**However, the activity of the ribozyme was not tested when exposed to vesicles without encapsulation, so it is unknown whether the observed effect was** due to greater denaturation or to a smaller encapsulation effect..." *(page 10)*

- Authors hypothesize that crowding phenomena can explain the chaperoning effect of liposomes on RNA. It would be very valuable to see experiments done with liposomes of many different sizes, as the crowding effect in the lumen will be vastly different in 100nm vesicles vs 1 or 2um ones.

Reply 14. Please see **Reply 4**.

Minor point:

- Figure 5 is purely speculative, it's not real data. It does not bring anything new to the narrative. If it's meant to be a concept figure, then figure 1 is redundant.

Reply 15. We deleted this figure.

- a lot of the paper, and most of the SI information, is focused on characterizing previously well-characterized systems, particularly on detailed characterization of MG aptamer. It is unclear what is new in this work, or why was this needed.

Reply 16. We have reduced the section in Results that confirmed previous work (old section

"Myristoleic acid (MA) vesicles" (see below for deleted text). Old Supplemental Figures 1, 2, and 5 (DLS of vesicles) and old Supplemental Figure 3A (fluorescence spectrum of bound and unbound MG in aqueous buffer) were removed. However, we retain two sentences describing DLS results since the vesicle size is used in calculations in the manuscript. We retained old Supplemental Figure 3B (binding curve of MG to the aptamer) and Supplemental Figure 6 (melting curve of the aptamer in aqueous buffer) because we found that the K_D is somewhat altered by the vesicle-compatible buffer used here, compared to previous work. Chromatograms of vesicle purification are shown as Supplementary Figures in response to comments of other reviewers. All remaining data in the Supplement and main text represent new experiments and results. In terms of main figures, we moved some figures to illustrate points that appear to be more important in light of the reviews (e.g., added K_D curves using MA:GMM and MA:MAOH, added a CD melting curve, and moved two figures regarding crowding agents from the supplement to the main figures).

- a) The following text was deleted from Results: "~~Fatty acid vesicles aggregate in the presence of relatively low concentrations of $MgCl_2$, which is often required for the activity of ribozymes or aptamers⁴². One solution to this problem is partial chelation of Mg^{2+} by citrate, which also increases the stability of encapsulated single-stranded RNA to degradation⁴³. We verified the stability of the vesicles in the presence of 1 mM Mg^{2+} -citrate by DLS (Supplementary Fig. 2).~~"
- b) Please note that the references in the deleted text are still cited in Methods (Encapsulation of RNA).

- when estimating magnesium level in presence of fatty acid liposomes, was the chelating effect of lipids included in calculating the available free magnesium?

Reply 17. The activity of magnesium inside vesicles was measured by mag-fura-2 fluorescence, and was found to not be significantly altered compared to the bulk volume. This experiment assays the magnesium available to bind mag-fura-2, which has a magnesium dissociation constant of 1.5 mM. It is possible that mag-fura-2 outcompetes weaker interactions with the fatty acids, which cannot be detected. We mention this caveat in the Discussion (see **Reply 3a**, last line). Note that the MG aptamer is relatively insensitive to magnesium concentration (ref 47), and the shift in K_D does not depend on the presence of Mg^{2+} (Figure 6c), so altered Mg^{2+} concentration affecting the aptamer is an unlikely mechanism for the observed effect (as mentioned in the Discussion regarding mechanism).

- what kind of lipids were used to make vesicles on Figure 3?

Reply 18. These were MA vesicles, which is now specified in the caption (now Figure 4a due to rearrangement of figures).

- when measuring K_d of ligand in presence of liposomes, it would be important to control for the affinity of the ligand to the bilayer membrane. Malachite green is hydrophobic and it would be easy to see how it binds to, or embeds, into hydrophobic lipid bilayer, which would shift the apparent measured K_d 's in presence of liposomes. I would suggest malachite green leakage assay, or co-purification (mix with empty liposomes, purify to see if malachite green elutes with

liposome fraction).

Reply 19. Please see **Reply 9.**

REVIEWERS' COMMENTS:

Reviewer #1 (Remarks to the Author):

In this manuscript, Chen and coworkers investigated that the stability of RNA aptamer in the inside fatty acid vesicles and outside the vesicles in the presence of cosolutes. This manuscript has been improved and they have taken the referee's comments. The author explain experimental procedure in more detail and carried out additional experiments. The discussion for effects of vesicles is well done and clearly supported their conclusion. However, this referee feels that the author should discuss below to much improve this manuscript.

In reply 5, the author concluded that the excluded volume effect in the vesicles can be separated from cosolute effects by the comparison between encapsulated RNA and RNA exposed to the same solution. The author should discuss what is the most important factor of determining aptamer stability. Moreover, the biological significance in cell for the stabilization mechanism of the aptamer stability change inside and outside vesicles should be explained.

Reviewer #2 (Remarks to the Author):

The authors adequately addressed all of my concerns. I support the publication of this work.

Reviewer #3 (Remarks to the Author):

The authors sufficiently addressed all my comments by detailed explanations and more experiments.

May 16, 2018

Response to Reviewer Comments

Reviewer #1 (Remarks to the Author):

In this manuscript, Chen and coworkers investigated that the stability of RNA aptamer in the inside fatty acid vesicles and outside the vesicles in the presence of cosolutes. This manuscript has been improved and they have taken the referee's comments. The author explain experimental procedure in more detail and carried out additional experiments. The discussion for effects of vesicles is well done and clearly supported their conclusion. However, this referee feels that the author should discuss below to much improve this manuscript.

In reply 5, the author concluded that the excluded volume effect in the vesicles can be separated from cosolute effects by the comparison between encapsulated RNA and RNA exposed to the same solution. The author should discuss what is the most important factor of determining aptamer stability.

We have added to the Discussion about this point, as follows (new text in bold font):

- "These results indicate that the excluded volume effect is not necessarily the primary driver of the effect of crowding agents on the MG aptamer, and chemical interactions with cosolutes would be an important factor affecting aptamer affinity. **As seen by the range of effects we observe in the presence of different crowding agents and lipids (Supplementary Fig. 7 and Supplementary Table 1), cosolutes can have a major effect on the affinity of the MG aptamer. The largest effect observed here was a 20-fold decrease in affinity upon exposure to POPC, and indeed positively charged species can cause nonspecific precipitation and aggregation of RNA. Thus chemical interactions with the RNA could be a dominant influence on aptamer affinity, depending on the solution composition.** However, unlike the excluded volume effect of crowding agents, the confinement effect of vesicles can be studied separately from the chemical interactions by experimental comparison with non-encapsulated RNA that is exposed to vesicles (RNA outside vesicles). This comparison indicates a similar increase in affinity (~**3-fold**) observed for the five lipid compositions tested here. " (page 12)

Moreover, the biological significance in cell for the stabilization mechanism of the aptamer stability change inside and outside vesicles should be explained.

We have added to the last paragraph of the Discussion, as follows.

- "An important caveat is that the effect of encapsulation was only studied for one aptamer here, in a limited number of chemical conditions and vesicle compositions. **In addition, while this biophysical effect may occur in a modern biological context, it is unclear whether it would be quantitatively important since chemical interactions with the complex cellular or exosomal contents may play a large role in determining RNA folding.** Further work would be needed to assess the generality of the effect. " (page 14)